# A Direct Second-Order Method for Solving Two-Player Zero-Sum Games

**David Yang** [1]  **Yuan Gao** [2]  **Tianyi Lin** [1]  **Christian Kroer** [1]

## Abstract

We introduce, to our knowledge, the first direct second-order method for computing Nash equilibria in two-player zero-sum games. To do so, we construct a Douglas-Rachford-style splitting formulation, which we then solve with a semismooth Newton (SSN) method. We show that our algorithm enjoys local superlinear convergence. To augment the fast local behavior of our SSN method with global efficiency guarantees, we develop a hybrid method that combines our SSN method with the state-of-the-art first-order method for game solving, Predictive Regret Matching$^+$ (PRM$^+$). Our hybrid algorithm leverages the global progress provided by PRM$^+$, while achieving a local superlinear convergence rate once it switches to SSN near a Nash equilibrium. Numerical experiments on matrix games demonstrate order-of-magnitude speedups over PRM$^+$ for high-precision solutions.

## 1. Introduction to Bilinear Saddle Point Problem

The Nash equilibrium is arguably the most foundational solution concept of game theory, representing a strategy profile in which no player can unilaterally deviate to improve their outcome. Approximating Nash equilibria of large-scale game models is the foundation of many human and superhuman-level AIs for games such as poker (Bowling et al., 2015; Brown & Sandholm, 2018; 2019), Stratego (Perolat et al., 2022), and Diplomacy (Meta Fundamental AI Research Diplomacy Team (FAIR) et al., 2022).

In the two-player zero-sum game setting, one player's gain is the other player's loss, so the game is completely described by a single payoff (utility) matrix $A \in \mathbb{R}^{n \times m}$, where entry $a_{ij}$ specifies the utility of the column player (and the negative utility of the row player) when the actions $i$ and $j$ are played. A (mixed-strategy) Nash equilibrium is then a pair of probability distributions $(x^*, y^*)$ over the rows and columns of $A$, such that $x^*$ and $y^*$ minimize and maximize their expected payoffs, respectively, given the strategy of the other player.

By von Neumann's minimax theorem, finding a Nash equilibrium in a two-player zero-sum game can be formulated as solving the bilinear saddle point problem (BLSP)

$$\min_{x \in X} \max_{y \in Y} x^\top A y \qquad (1)$$

where $A \in \mathbb{R}^{n \times m}$ is the payoff matrix and $X \subseteq \mathbb{R}^n$, $Y \subseteq \mathbb{R}^m$ are convex, compact strategy sets for the min and max players.

In both normal-form and extensive-form games, $X$ and $Y$ are polytopes, and so the optimization problem in Equation (1) can be solved exactly by linear programming (Von Stengel, 1996). While this approach works in principle, it is impractical for large-scale games; in that setting, with state-of-the-art commercial solvers, the LP reformulation inflates the problem size and removes exploitable structures in the payoff and constraint matrices, making exact solutions computationally prohibitive.

Instead, first-order methods (FOMs) and regret minimization approaches have been widely used for large-scale game solving, where they have been highly successful in scaling to and solving large games arising from real-world applications. FOMs such as the *Extragradient Method* (Korpelevich, 1976) and *Mirror Prox* (Nemirovski, 2004) converge to a Nash equilibrium at a rate of $O(1/T)$, where $T$ is the number of iterations. Regret minimization approaches include *Regret Matching$^+$* (RM$^+$) and its predictive variant *Predictive Regret Matching$^+$* (PRM$^+$) (Zinkevich et al., 2007); these approaches use self-play to iteratively update each player's strategy proportional to their positive cumulative regrets, and converge to a Nash equilibrium at a rate of $O(1/\sqrt{T})$, though their empirical performance is often much stronger. In fact, *Counterfactual Regret Minimization$^+$* (CFR$^+$) (Tammelin, 2014) and its predictive variant PCFR$^+$ (Farina et al., 2021), which extend the RM$^+$ framework from normal-form to extensive-form games, are some of the fastest game-solving algorithms numerically, and CFR$^+$ served as the foundation for recent breakthroughs

---

[1]Columbia University, New York, NY, USA [2]Microsoft, Redmond, WA, USA. Correspondence to: David Yang <dy2507@columbia.edu>.

*Proceedings of the 43$^{rd}$ International Conference on Machine Learning*, Seoul, South Korea. PMLR 306, 2026. Copyright 2026 by the author(s).

in superhuman poker AIs (Bowling et al., 2015; Brown & Sandholm, 2018; 2019).

While FOMs serve as the standard for computing approximate Nash equilibrium in two-player zero-sum games, their sublinear convergence guarantees make high-precision solutions expensive to compute. In convex minimization, high-precision solutions can be computed with second-order methods, which exploit curvature information for rapid local convergence. In two-player zero-sum game solving, however, there are no known "direct" methods for computing Nash equilibria using second-order methods. Classical second-order methods in optimization, such as interior-point algorithms, exploit curvature information through barrier formulations to achieve polynomial-time complexity for constrained problems. These methods can be applied by converting the BLSP into a linear program and applying the interior-point methods to that formulation. However, this throws away much of the structure of the BLSP and forces the use of generic methodology.

Recently, there have been advances in applying second-order methods directly to saddle point problems, such as the $J$-symmetric quasi-Newton framework of Asl et al. (2024), but these advances are restricted to the unconstrained setting. In contrast, our method offers a direct means of incorporating curvature information into bilinear saddle point problems, without resorting to barrier reformulations or unconstrained analogues, while simultaneously leveraging the strengths of the state-of-the-art FOMs in game solving.

As a first step to leverage the power of the curvature information inherent in the strategy spaces, we present a hybrid method combining a state-of-the-art FOM with a semi-smooth Newton method applied to an operator stemming from the Douglas-Rachford splitting algorithm (DRS). Our main results are as follows.

**Principled Use of Curvature Information.** To the best of our knowledge, we develop the first direct second-order method for solving bilinear saddle point problems, based on a Douglas–Rachford splitting formulation solved via a regularized semi-smooth Newton method. In contrast to other saddle point residual operators, our residual operator derived from DRS is monotone and piecewise affine, making it particularly well-suited for SSN methods. We establish that the regularized semi-smooth Newton method applied to our residual operator achieves local quadratic convergence. Our analysis replaces the classical BD-regularity assumption (Facchinei & Pang, 2003) with a local error bound condition and a stability property implied by monotonicity and Lipschitz continuity. In doing so, we extend prior metric-subregularity results for matrix games (Tseng, 1995; Gilpin et al., 2012; Wei et al., 2021) to our residual operator using tools from convex analysis.

**First and Second-order Hybrid Method.** While our second-order method has local superlinear convergence, it may be slow initially. To ameliorate this fact, we develop a principled approach for switching between iterates generated by a first-order method, which generally lie in the feasible set, and iterates for our SSN method, which generally move outside the feasible set. By systematically leveraging the progress of PRM$^+$ to warm-start our regularized semi-smooth Newton method, our approach provides the first framework combining first-order progress with second-order guarantees, ensuring efficient global progress together with local superlinear convergence as our method transitions to SSN steps. A key component of our design is a lifting framework that allows for an effective transition: given a strong FOM iterate, the corresponding lifted SSN iterate has a small residual norm, placing it squarely within the regime of local superlinear convergence.

**Numerical Experiments.** We conduct extensive numerical experiments on a diverse set of matrix games. For finding low-precision regimes, our hybrid method can be slower than pure FOMs due to a preprocessing inversion step and the overhead of solving Newton systems. However, once the SSN iterates enter a neighborhood of a Nash equilibrium in the lifted space, our hybrid method effectively exploits second-order information, leading to significant speedup over state-of-the-art FOMs. The results validate both the theoretical guarantees and the practical efficiency of our approach for high-precision equilibrium computation.

## 2. Preliminaries and Technical Background

We provide the formal definitions for two-player zero-sum games considered in this paper, and the schemes of operator splitting methods and semi-smooth Newton methods.

### 2.1. Two-Player Zero-Sum Games

We denote by $\mathcal{S} = \Delta^n \times \Delta^m$ the set of feasible strategy profiles, where $\Delta^n$ and $\Delta^m$ are the simplices representing the mixed strategies of the two players in an $n \times m$ zero-sum matrix game with payoff matrix $A$. A generic strategy profile will be written as $z := (x, y)$, with $x \in \Delta^n$ representing the minimizer's strategy and $y \in \Delta^m$ the maximizer's strategy. Within this framework, the duality gap serves as a natural measure of how far a strategy profile is from the set of Nash equilibria: it quantifies the incentive for either player to unilaterally deviate from their current strategies.

**Definition 2.1.** For a strategy profile $\hat{z} = (\hat{x}, \hat{y})$, the duality gap is defined as

$$\text{gap}(\hat{z}) = \max_{y \in \Delta^m} \hat{x}^\top A y - \min_{x \in \Delta^n} x^\top A \hat{y}. \qquad (2)$$

By definition, $\hat{z}$ is a Nash equilibrium if and only if $\text{gap}(\hat{z}) = 0$.

The maximizer evaluates the highest payoff achievable by deviating from $\hat{y}$, while the minimizer evaluates the lowest payoff achievable by deviating from $\hat{x}$, demonstrating that $\text{gap}(\hat{z}) \geq 0$ always holds. In practice, iterative algorithms for computing Nash equilibria seldom converge exactly to an equilibrium, but rather produce approximate solutions. This motivates the notion of an $\epsilon$-Nash equilibrium, where the incentives to deviate are bounded by a small tolerance $\epsilon > 0$.

**Definition 2.2.** A strategy profile $\hat{z} = (\hat{x}, \hat{y})$ is an $\epsilon$-Nash equilibrium if the following condition holds:

$$\max_{y \in \Delta^m} \hat{x}^\top A y - \epsilon \leq \hat{x}^\top A \hat{y} \leq \min_{x \in \Delta^n} x^\top A \hat{y} + \epsilon.$$

*Remark* 2.3. The concept of an $\epsilon$-Nash equilibrium can be equivalently expressed in terms of the duality gap (Orabona, 2019). Specifically, if $\hat{z}$ is an $\epsilon$-Nash equilibrium, then its duality gap satisfies $\text{gap}(\hat{z}) \leq 2\epsilon$. Conversely, if a strategy profile $\hat{z}$ achieves a duality gap below $\epsilon$, then $\hat{z}$ itself constitutes an $\epsilon$-Nash equilibrium. This tight connection establishes the duality gap as a natural and widely used performance metric for analyzing iterative algorithms.

## 2.2. Operator Splitting Methods

As an alternative to the duality gap, Nash equilibria can also be characterized through residual operators. A natural choice is the classical saddle-point normal map $z \mapsto z - \Pi_{\mathcal{S}}\left(z - \eta \begin{bmatrix} 0 & A \\ -A^\top & 0 \end{bmatrix} z\right)$, but this map is not monotone in the two-player zero-sum game setting and therefore lacks a desired structural property for SSN methods. Instead, we base our construction on the optimality condition for two-player zero-sum games: a strategy profile $\hat{z} = (\hat{x}, \hat{y}) \in \mathcal{S}$ is a Nash equilibrium if and only if $0 \in F(\hat{z}) + N_{\mathcal{S}}(\hat{z})$ where $F(\hat{z}) = \begin{bmatrix} 0 & A \\ -A^\top & 0 \end{bmatrix} \hat{z}$ and $N_{\mathcal{S}}(\hat{z}) = \{v \mid v^\top(z - \hat{z}) \leq 0 \quad \forall z \in \mathcal{S}\}$.

Thus, finding a Nash equilibrium reduces to finding a root of the sum of two structured operators. From standard convex analysis, both $F(\cdot)$ and $N_{\mathcal{S}}(\cdot)$ are maximally monotone, which guarantees that their resolvents $J_F^\gamma(\cdot) = (I + \gamma F)^{-1}(\cdot)$ and $J_{N_{\mathcal{S}}}^\gamma = (I + \gamma N_{\mathcal{S}})^{-1}(\cdot)$ are single-valued and firmly nonexpansive. This fact underpins the use of operator-splitting methods in solving two-player zero-sum games; for an overview of such methods, see the monographs Rockafellar & Wets (1998); Bauschke & Combettes (2017); Ryu & Yin (2022).

In particular, we focus on the Douglas-Rachford splitting (DRS) method (Lions & Mercier, 1979; Eckstein & Bertsekas, 1992), which applies to sums of two maximal monotone operators. With step size $\gamma > 0$, the DRS operator

applied to our setting is given by

$$T_{\text{DRS}}^\gamma = \text{Id} - J_{N_{\mathcal{S}}}^\gamma + J_F^\gamma \circ (2J_{N_{\mathcal{S}}}^\gamma - \text{Id}), \qquad (3)$$

where $\text{Id}$ denotes the identity operator and $\circ$ represents composition. Because $F(\cdot)$ is linear and $N_{\mathcal{S}}(\cdot)$ is a normal cone operator, their resolvents simplify to

$$J_F^\gamma(z) = \begin{bmatrix} I & \gamma A \\ -\gamma A^\top & I \end{bmatrix}^{-1} z, \quad J_{N_{\mathcal{S}}}^\gamma(z) = \Pi_{\mathcal{S}}(z).$$

Thus, $J_F^\gamma$ is a linear operator and $J_{N_{\mathcal{S}}}^\gamma$ is simply the projection onto $\mathcal{S}$. As a consequence, one DRS iteration takes the following form of

$$\begin{aligned} z_{k+1} &= T_{\text{DRS}}^\gamma(z_k) \\ &= z_k - \Pi_{\mathcal{S}}(z_k) + \begin{bmatrix} I & \gamma A \\ -\gamma A^\top & I \end{bmatrix}^{-1} (2\Pi_{\mathcal{S}}(z_k) - z_k). \end{aligned}$$

$$(4)$$

We define the residual operator $R_{\text{DRS}}^\gamma := \text{Id} - T_{\text{DRS}}^\gamma$, noting that $R_{\text{DRS}}^\gamma(\hat{z}) = 0$ implies $\Pi_{\mathcal{S}}(\hat{z})$ is a Nash equilibrium (see Section 3.1). The following lemma summarizes one of its key properties.

**Lemma 2.4.** *The operator $R_{\text{DRS}}^\gamma(\cdot)$ is*

*(i) monotone:* $(z_1 - z_2)^\top (R_{\text{DRS}}^\gamma(z_1) - R_{\text{DRS}}^\gamma(z_2)) \geq 0$

*(ii) 1-Lipschitz:* $\|R_{\text{DRS}}^\gamma(z_1) - R_{\text{DRS}}^\gamma(z_2)\| \leq \|z_1 - z_2\|$

*for all $z_1, z_2$.*

Our approach is to solve the root-finding problem for this residual operator using semi-smooth Newton (SSN) methods (Qi & Sun, 1993; Pang & Qi, 1993; Martínez & Qi, 1995). This shares a similar spirit with prior SSN applications in composite convex optimization (Xiao et al., 2018) and kernel-based optimal transport (Lin et al., 2024). However, the unique structure of two-player zero-sum games requires new algorithmic strategies, including nontrivial switching between PRM$^+$ and SSN, making the practical implementation considerably more delicate.

## 2.3. Semi-smooth Newton Methods

Semi-smooth Newton (SSN) methods generalize the classical Newton method to nonsmooth equations by replacing the Jacobian with a generalized Jacobian. Since $R_{\text{DRS}}^\gamma(\cdot)$ is Lipschitz continuous, Rademacher's theorem implies that it is differentiable almost everywhere. This motivates the use of the notion of a generalized Jacobian (Clarke, 1990).

**Definition 2.5.** Let $F$ be a Lipschitz mapping and $D_F$ the set of differentiable points of $F$. The $B$-subdifferential at $z$ is $\partial_B F(z) = \{\lim_{k \to +\infty} \nabla F(z^k) \mid z^k \in D_F, z^k \to z\}$ and the generalized Jacobian is defined as $\partial F(z) = \text{conv}(\partial_B F(z))$, where conv denotes the convex hull.

A second key notion is semi-smoothness, introduced by Mif-flin (1977) for real-valued functions and extended to vector-valued mappings by Qi & Sun (1993).

**Definition 2.6.** Let $F$ be a Lipschitz mapping. Then $F$ is (strongly) semi-smooth at $z$ if (i) $F$ is directionally differentiable at $z$, and (ii) for any $G \in \partial F(z + \Delta z)$, as $\Delta z \to 0$,

$$\text{(semismooth)} \quad \frac{\|F(z+\Delta z)-F(z)-G\Delta z\|}{\|\Delta z\|} \to 0,$$
$$\text{(strongly semismooth)} \quad \frac{\|F(z+\Delta z)-F(z)-G\Delta z\|}{\|\Delta z\|^2} \leq C$$

for some constant $C > 0$.

The following lemma shows that our residual operator $R_{\mathrm{DRS}}^\gamma(\cdot)$ is piecewise affine and strongly semi-smooth, which justifies the use of SSN to solve the root-finding problem $R_{\mathrm{DRS}}^\gamma(z) = 0$.

**Lemma 2.7.** *The operator $R_{\mathrm{DRS}}^\gamma(\cdot)$ is piecewise affine and strongly semi-smooth.*

Our regularized SSN method will proceed as follows: given $z_k$, compute $z_{k+1} = z_k + \Delta z_k$, where $\Delta z_k$ solves

$$(G_k + \mu_k I)\Delta z_k = -r_k, \tag{5}$$

with $G_k \in \partial R_{\mathrm{DRS}}^\gamma(z_k)$, $r_k = R_{\mathrm{DRS}}^\gamma(z_k)$, and $I$ the identity matrix. The regularization parameter is chosen adaptively as $\mu_k = \theta_k \|r_k\|$ for $\theta_k > 0$. This stabilizes the method in practice: as $r_k \to 0$, the algorithm gradually exploits curvature information from $G_k$ while maintaining invertibility of $G_k + \mu_k I$.

If $R$ is continuously differentiable and $\theta_k = 0$, this reduces to the Newton method, which exhibits local quadratic convergence. Although regularized SSN methods may diverge in general (Kummer, 1988), local superlinear convergence is guaranteed under strong semi-smoothness and local error bound conditions (Zhou & Toh, 2005), and we establish these conditions in Section 3.3.

## 3. Main Results

We first establish a formal connection between Nash equilibria and lifted fixed points, which serves as the theoretical foundation of our approach. Second, building on this insight, we introduce a new game-solving algorithm that integrates PRM$^+$ with a regularized semi-smooth Newton method. Finally, we provide a convergence analysis of the resulting procedure.

### 3.1. Linking Nash Equilibria to Lifted Fixed Points

While Nash equilibria are not themselves fixed points of $T_{\mathrm{DRS}}^\gamma$, they can be linked through a simple "lift", as the following theorem shows.

**Theorem 3.1.** *The following relationships between fixed points of $T_{\mathrm{DRS}}^\gamma$ and Nash equilibria hold:*

(i) *If $z \in \mathbb{R}^{m+n}$ is a fixed point of $T_{\mathrm{DRS}}^\gamma$, then $\hat{z} = \Pi_\mathcal{S}(z)$ is a Nash equilibrium.*

(ii) *If $\hat{z} \in \mathcal{S}$ is a Nash equilibrium, then the lifted point $z = \hat{z} - \gamma F(\hat{z})$ is a fixed point of $T_{\mathrm{DRS}}^\gamma$.*

*Remark* 3.2. A Nash equilibrium is not necessarily a fixed point of $T_{\mathrm{DRS}}^\gamma$. Indeed, since $\Pi_\mathcal{S}(\hat{z}) = \hat{z}$, we have

$$T_{\mathrm{DRS}}^\gamma(\hat{z}) = \hat{z} - \Pi_\mathcal{S}(\hat{z}) + \begin{bmatrix} I & \gamma A \\ -\gamma A^\top & I \end{bmatrix}^{-1} (2\Pi_\mathcal{S}(\hat{z}) - \hat{z})$$

$$= \begin{bmatrix} I & \gamma A \\ -\gamma A^\top & I \end{bmatrix}^{-1} \hat{z}.$$

Thus, $\hat{z}$ is a fixed point of $T_{\mathrm{DRS}}^\gamma$ if and only if $F(\hat{z}) = 0$, which is a strictly stronger condition than the first-order optimality condition for Nash equilibrium. The role of the lifted point $z = \hat{z} - \gamma F(\hat{z})$ is to displace $\hat{z}$ along its normal cone direction, which ensures the fixed-point relation.

To implement the SSN method, we require an efficient way to obtain one element in $\partial R_{\mathrm{DRS}}^\gamma$. By definition, we have

$$R_{\mathrm{DRS}}^\gamma = J_{N_\mathcal{S}}^\gamma - J_F^\gamma \circ (2J_{N_\mathcal{S}}^\gamma - \mathrm{Id}),$$

where

$$J_F^\gamma(z) = \begin{bmatrix} I & \gamma A \\ -\gamma A^\top & I \end{bmatrix}^{-1} z, \quad J_{N_\mathcal{S}}^\gamma(z) = \Pi_\mathcal{S}(z).$$

The generalized Jacobian of $R_{\mathrm{DRS}}^\gamma(\cdot)$ at $z$ is given by

$$\partial R_{\mathrm{DRS}}^\gamma(z) = \partial J_{N_\mathcal{S}}^\gamma(z)$$
$$- (\partial J_F^\gamma) \circ (2\partial J_{N_\mathcal{S}}^\gamma(z) - z) \cdot (2\partial J_{N_\mathcal{S}}^\gamma(z) - I),$$

where

$$\partial J_F^\gamma(z) = \begin{bmatrix} I & \gamma A \\ -\gamma A^\top & I \end{bmatrix}^{-1}$$

and

$$\partial J_{N_\mathcal{S}}(z) = \partial \Pi_\mathcal{S}(z)$$
$$= \left\{ \begin{bmatrix} G_x & 0 \\ 0 & G_y \end{bmatrix} : G_x \in \partial \Pi_{\Delta^n}(x), G_y \in \partial \Pi_{\Delta^m}(y) \right\}.$$

Thus, finding an element in $\partial R_{\mathrm{DRS}}^\gamma$ reduces to finding an element of the generalized Jacobian of the simplex projection operator.

**Theorem 3.3.** *Let $p \in \mathbb{R}^d$ and let $x = \Pi_{\Delta^d}(p)$ denote the projection of $p$ onto the simplex $\Delta^d$. Define the active set $\mathcal{A} := \{i \in [d] : x_i > 0\}$. Then*

$$G = \mathrm{diag}(a) - \frac{1}{\|a\|_1} aa^\top,$$

*where $a \in \mathbb{R}^d$ is the indicator vector of the active set $\mathcal{A}$, satisfies $G \in \partial \Pi_{\Delta^d}(p)$.*

Putting these pieces together yields

$$\begin{bmatrix} G_x & 0 \\ 0 & G_y \end{bmatrix} - \begin{bmatrix} I & \gamma A \\ -\gamma A^\top & I \end{bmatrix}^{-1} \left( 2 \begin{bmatrix} G_x & 0 \\ 0 & G_y \end{bmatrix} - I \right)$$
$$\in \partial R_{\mathrm{DRS}}^\gamma(z).$$

(6)

Theorem 3.1 establishes that projecting a fixed point of $T_{\mathrm{DRS}}^\gamma$ onto $\mathcal{S}$ yields a Nash equilibrium, and conversely, every Nash equilibrium induces a lifted fixed point. These results establish a correspondence between exact fixed points and exact Nash equilibria through projection and lifting.

In practice, however, algorithms rarely reach exact solutions; instead, they operate with approximate fixed points and $\epsilon$-Nash equilibria. We therefore extend this correspondence to the approximate setting. In particular, we show how the residual norm of $R_{\mathrm{DRS}}^\gamma(\cdot)$ can be related to the duality gap. This link guarantees that progress made by a first-order method (which decreases the duality gap) translates into progress for the semi-smooth Newton method (which works to decrease the residual norm), and vice versa. As such, the connection enables a principled transition between the two regimes within our proposed algorithmic framework.

To formalize this relationship, we require the following condition measure of the payoff matrix $A$, which is used to control the Euclidean distance to the set of equilibria in terms of the duality gap. Being able to relate the duality gap to the iterate distance serves as an essential tool in our analysis.

For the following discussion, we let $Z^*$ denote the set of zeros of $R_{\mathrm{DRS}}^\gamma(\cdot)$ and let $\hat{Z}^*$ denote the set of Nash equilibria of $\mathcal{S}$. For ease of reference, we also define $\mathrm{dist}(z, Z^*) := \min_{z^* \in Z^*} \|z - z^*\|$.

**Definition 3.4** (Condition Measure (Gilpin et al., 2012)). The condition measure of a matrix $A$, denoted $\delta(A)$, is defined as

$$\delta(A) = \sup \left\{ \delta : \mathrm{dist}(\hat{z}, \hat{Z}^\star) \leq \frac{\mathrm{gap}(\hat{z})}{\delta} \text{ for all } \hat{z} \in \mathcal{S} \right\}.$$

*Remark* 3.5. In addition, Gilpin et al. (2012, Lemma 4) guarantees that there exists $\delta > 0$ such that

$$\mathrm{dist}(\hat{z}, \hat{Z}^\star) \leq \frac{\mathrm{gap}(\hat{z})}{\delta} \tag{7}$$

for all $\hat{z} \in \mathcal{S}$. This implies $\delta(A) < +\infty$ for non-degenerate instances.

**Theorem 3.6.** *Let* $\hat{z} \in \mathcal{S}$. *The following relationships hold:*

*(i) If* $\|R_{\mathrm{DRS}}^\gamma(z)\| \geq \tau \mathrm{dist}(z, Z^*)$ *for some constant* $\tau > 0$, *we have*

$$\mathrm{gap}(\Pi_\mathcal{S}(z)) \leq \frac{\sqrt{2}\|A\| \cdot \|R_{\mathrm{DRS}}^\gamma(z)\|}{\tau}.$$

---

**Algorithm 1** Hybrid SSN method
1: **Input**: $p_0 = z_0 \in \Delta^n \times \Delta^m$, $\theta_0 > 0$, $\gamma > 0$
2: Calculate resolvent for DRSSN method
3: **for** $k = 0, 1, \ldots$ **do**
4:     Compute $p_{k+1}$ from $p_k$ using one-step of PRM$^+$.
5:     **if** ready to update $\theta_k$ **then**
6:         Update $\theta_k$ adaptively.
7:     **end if**
8:     **if** ready to switch to DRSSN method **then**
9:         Set $z_k = p_k - \gamma F(p_k)$.
10:         Run DRSSN$(z_k, \theta_k)$.
11:     **end if**
12: **end for**

---

*(ii) For the lifted point* $z = \hat{z} - \gamma F(\hat{z})$, *we have*

$$\|R_{\mathrm{DRS}}^\gamma(z)\| \leq \frac{(1 + \gamma\|A\|)\mathrm{gap}(\hat{z})}{\delta(A)}.$$

### 3.2. Algorithmic Framework

We now outline the general framework of our algorithm; we refer the reader to Appendix B for details. The core idea is to warm-start our Douglas-Rachford-based regularized SSN method (DRSSN) using iterates generated by a phase of exclusively PRM$^+$ steps. The PRM$^+$ phase reduces the duality gap to bring iterates sufficiently close to a Nash equilibrium, thereby avoiding the instability that can arise when DRSSN is initialized by lifting an iterate arbitrarily far from a Nash equilibrium and started in a regime where curvature information is not yet informative. Moreover, the steps in the PRM$^+$ phase can be used to update the damping parameter for DRSSN. After the duality gap drops below a prescribed threshold, our algorithm makes a one-time switch from PRM$^+$ to DRSSN, after which only Newton steps are taken. This design ensures global efficiency from the first-order phase and rapid local convergence from the second-order phase.

The DRSSN subroutine implements our regularized semi-smooth Newton method. Each iteration involves solving a linear system to compute the Newton step $\Delta z_k$, followed by a damping parameter update that depends on changes in the residual norm $\|R_{\mathrm{DRS}}^\gamma\|$. While the duality gap is the standard measure of proximity to a Nash equilibrium and is natural for first-order methods that remain within the feasible strategy spaces, it is less effective for guiding second-order methods. In particular, the SSN iterates may leave the feasible region, and relying solely on the projected duality gap may fail to capture true progress. Instead, effective performance of our SSN method hinges on proper tuning of the damping parameter to control the residual norm, which in turn governs its convergence.

This distinction is crucial in our hybrid algorithm. The first-

**Algorithm 2** DRSSN($z, \lambda$)

1: **Input**: $z_0 = z \in \mathbb{R}^{n+m}$, $\lambda_0 = \lambda > 0$, $\ell = 1.5$
2: **for** $k$ in $0, 1, \ldots,$ **do**
3:     **while** $\lambda_k \leq 10^9$ **do**
4:         Select $G_k \in \partial R^\gamma_{\mathrm{DRS}}(z_k)$.
5:         Use $G_k$ and solve the linear system in
        Equation (5) for $\Delta z_k$.
6:         Define $z' := z_k + \Delta z_k$.
7:         **if** $\|R^\gamma_{\mathrm{DRS}}(z')\| < \|R^\gamma_{\mathrm{DRS}}(z_k)\|$ **then**
8:             Set $z_{k+1} = z'$.
9:             Update $\lambda_{k+1} = \lambda_k / \ell$.
10:             **break**
11:         **else**
12:             Update $\lambda_{k+1} = \lambda_k \cdot \ell$.
13:         **end if**
14:     **end while**
15:     Update $\lambda_{k+1}$ using the adaptive scheme.
16: **end for**
17: **if** gap($\Pi_\mathcal{S}(z_{k+1})$) is under desired gap **then**
18:     **return** $\Pi_\mathcal{S}(z_{k+1})$.
19: **end if**

order phase reduces the duality gap to a chosen threshold, ensuring a stable lifted iterate for the second-order phase. In particular, if the duality gap of the PRM$^+$ iterates is sufficiently small, then by Theorem 3.6, the residual norm of the corresponding lifted point is appropriately bounded, ensuring that it lies within the regime where our SSN theory guarantees local superlinear convergence. The second-order phase advances through Newton steps while adaptively adjusting the damping parameter based on the residual norm, thereby exploiting curvature information to achieve accelerated convergence. To this end, our SSN method employs two damping-update strategies: (i) a line-search regime, which adjusts the parameter according to whether the residual norm decreases, thus ensuring consistent global progress when iterates are far from equilibrium; and (ii) an adaptive regime, inspired by Lin et al. (2024), which modifies the parameter heuristically to stabilize performance when the line-search regime stalls. Further details on the adaptive scheme are provided in Appendix B.

### 3.3. Convergence Guarantee

We begin by analyzing the local behavior of the second-order phase (Algorithm 2) of our hybrid method. Since the SSN steps of our hybrid algorithm are applied to the residual operator $R^\gamma_{\mathrm{DRS}}(\cdot)$, we first outline several structural properties that enable a local superlinear convergence guarantee.

Our first two properties are that $R^\gamma_{\mathrm{DRS}}(\cdot)$ is monotone and Lipschitz continuous (Lemma 2.4). These properties ensure

that every choice $G \in \partial R^\gamma_{\mathrm{DRS}}(z)$ in an SSN step will yield a positive semidefinite matrix (Xiao et al., 2018, Lemma 3.5). The following stability condition is then immediately satisfied (Zhou & Toh, 2005, Lemma 3.1).

**Lemma 3.7.** *For $G \in \partial R^\gamma_{\mathrm{DRS}}(z)$ and any $\mu > 0$,*

$$\|(G + \mu I)^{-1}\| \leq \frac{1}{\mu}.$$

Next, from Lemma 2.7, $R^\gamma_{\mathrm{DRS}}(\cdot)$ is strongly semi-smooth and piecewise affine. The following condition is motivated by rewriting these two properties into a quadratic bound, where $N(z^*)$ denotes a neighborhood of a point $z^* \in Z^*$. We additionally emphasize that the constant $L_2$ must be uniform over all $z^* \in Z^*$. Although we do not establish such uniformity theoretically, our numerical experiments exhibit behavior consistent with the existence of a uniform bound.[1]

**Lemma 3.8.** *For any $z^* \in Z^*$, there exists a neighborhood $N(z^*)$ and a constant $L_2 > 0$ such that for all $z \in N(z^*)$,*

$$\|R^\gamma_{\mathrm{DRS}}(z) - G(z - z^*)\| \leq L_2 \|z - z^*\|^2$$

*for any $G \in \partial R^\gamma_{\mathrm{DRS}}(z)$.*

The following two-point quadratic bound is similarly inspired by the combination of the strongly semi-smooth and piecewise affine property of our operator $R^\gamma_{\mathrm{DRS}}(\cdot)$.

**Lemma 3.9.** *There exists a neighborhood $N(Z^*)$ of the solution set $Z^*$ and a constant $L_3 > 0$ such that for all $z_1, z_2 \in N(Z^*)$,*

$$\|R^\gamma_{\mathrm{DRS}}(z_1) - R^\gamma_{\mathrm{DRS}}(z_2) - G(z_1 - z_2)\| \leq L_3 \|z_1 - z_2\|^2$$

*for some $G \in \partial R^\gamma_{\mathrm{DRS}}(z_2)$.*

We note that the above condition is only required to hold for *some* $G \in \partial R^\gamma_{\mathrm{DRS}}(z_2)$, rather than for *all* choices of $G \in \partial R^\gamma_{\mathrm{DRS}}(z_1)$, the latter of which follows from standard semi-smooth definitions alone. In our setting, we only require the choice of the generalized Jacobian element (e.g. via Theorem 3.3) to satisfy the quadratic bound. A key property underlying this result is the piecewise affine structure of the operator $R^\gamma_{\mathrm{DRS}}$: intuitively, once the iterates are sufficiently close to a solution, they lie in the interior of a region on which $R^\gamma_{\mathrm{DRS}}$ is affine, which directly yields the desired quadratic bound.

Finally, we require the following local error-bound condition which holds because $R^\gamma_{\mathrm{DRS}}(\cdot)$ is a piecewise affine (and thus polyhedral) mapping.

---

[1]More generally, we also expect that a local superlinear convergence result can still be established without this assumption by adapting arguments similar to those of Zhou & Toh (2005).

**Lemma 3.10.** *There exists a neighborhood $N(Z^*)$ of $Z^*$ and a constant $\tau > 0$ such that for all $z \in N(Z^*)$,*

$$\|R_{\mathrm{DRS}}^\gamma(z)\| \geq \tau \operatorname{dist}(z, Z^*).$$

This local error-bound condition is also known as local metric subregularity. In the context of matrix games, both the duality gap operator (Gilpin et al., 2012) and the gradient operator $F(z)$ (Wei et al., 2021) have been shown to satisfy variants of this property. In our setting, the residual operator is piecewise affine and thus polyhedral, which by Robinson (1981) implies local metric subregularity. Thus, we establish a direct extension of prior results and show that the analysis applies more generally to any polyhedral mapping.

These conditions help enforce the local quadratic convergence of the Newton steps of our hybrid algorithm.

**Theorem 3.11.** *The regularized semi-smooth Newton method defined by the update rule in Equation (5) applied to $R_{\mathrm{DRS}}^\gamma(\cdot)$ exhibits local quadratic convergence. Concretely, there exists a $k_0 \in \mathbb{N}$ such that the iterates $\{z_k\}_{k \geq k_0}$ satisfy*

$$\|R_{\mathrm{DRS}}^\gamma(z_{k+1})\| \leq C\|R_{\mathrm{DRS}}^\gamma(z_k)\|^2 \tag{8}$$

*where $C = \frac{1}{\tau^2}\left[L_3\left(2 + \frac{L_2}{\theta\tau}\right)^2 + \theta L_1\left(2 + \frac{L_2}{\theta\tau}\right)\right]$.*

*Remark* 3.12. Note that the above result does not rely on $\theta$ being fixed (see Appendix A.2). Consequently, the final bound in Equation (8) also holds in the case where each $\theta_k \in [\underline{\theta}, \overline{\theta}]$, i.e. bounded between fixed lower and upper bounds for $\theta$. This holds in many adaptive methods (Xiao et al., 2018; Lin et al., 2024), and ours, which vary $\theta$ at each time step.

While the above analysis establishes the local quadratic convergence of our residual operator, it applies more generally to any operator satisfying the stated conditions. Our analysis builds on the framework of Zhou & Toh (2005) and extends classical SSN theory by replacing the restrictive BD-regularity assumption (Xiao et al., 2018) with weaker and more broadly applicable conditions.

Now that we have established the local quadratic convergence of the SSN phase, we will connect the progress of the first-order method with entry into this local regime by presenting an end-to-end convergence characterization of our hybrid algorithm (Algorithm 1).

First, we require an additional instance-dependent parameter, $\epsilon(A, \gamma)$, beyond the condition measure $\delta(A)$ (Definition 3.4) and spectral norm $\|A\|$ of the payoff matrix. The quantity $\epsilon(A, \gamma)$ characterizes the size of the local region as measured by the norm of the operator $R_{\mathrm{DRS}}^\gamma$ at a point $z$. In particular, $\|R_{\mathrm{DRS}}^\gamma(z)\| \leq \epsilon(A, \gamma)$ will guarantee that we are within a local region in which the quadratic convergence governed by Equation (8) holds.

The following result quantifies the number of first-order iterations required before the lifted iterate enters a region in which the second-order phase exhibits quadratic convergence.

**Theorem 3.13** (End-to-end Convergence Rate of Algorithm 1)**.** *Define $\tilde{C} := \frac{\epsilon(A,\gamma)\,\delta(A)}{1+\gamma\,\|A\|}$. Suppose that the first-order phase[2] of Algorithm 1 generates iterates whose duality gap decreases at rate $O(1/T)$. Then, after at most $O(1/\tilde{C})$ first-order iterations, the corresponding lifted iterate enters a neighborhood in which the SSN phase is guaranteed to converge quadratically.*

*Proof.* Recall that notationally, $\hat{z}$ is an iterate produced by a first-order method. By Theorem 3.6 (ii), if

$$\|R_{\mathrm{DRS}}^\gamma(z)\| \leq \frac{(1+\gamma\|A\|)\mathrm{gap}(\hat{z})}{\delta(A)} \leq \epsilon(A, \gamma),$$

then the lifted point $z = \hat{z} - \gamma F(\hat{z})$ has residual norm bounded above by $\epsilon(A, \gamma)$, thus placing it in the region of local quadratic convergence by the definition of $\epsilon(A, \gamma)$. Rearranging, it follows that the lifted iterate enters the region of local quadratic convergence precisely when $\mathrm{gap}(\hat{z}) \leq C$; this happens after $O(1/C)$ iterations of the first-order method. $\square$

We note that Theorem 3.13 does not provide a computable switching rule in the absence of knowledge of the instance-dependent parameters. We present this result primarily to highlight that the theoretical guarantee of local superlinear convergence can be attained after a finite first-order phase. However, as we demonstrate in the following section, the region of fast local convergence is often quite large in practice: empirically, the SSN phase consistently exhibits rapid convergence even when the switch from the first-order phase is made at relatively coarse duality gap thresholds.

Finally, we note that the reliance on instance-dependent parameters for bounding local regions of convergence is not specific to our setting. Classical convergence analyses frequently depend on quantities such as the Hoffman constant for systems of linear inequalities (Hoffman, 1952) or condition measures arising in game-theoretic and optimization contexts (Mordukhovich et al., 2010). While these quantities play a fundamental theoretical role, they are notoriously difficult to compute or approximate in practice.

Theorem 3.13 formalizes the intuition underlying our hybrid design: the first-order phase reduces the duality gap until the corresponding lifted iterate can enter a neighborhood in which second-order information becomes reliable. At that point, Theorem 3.11 guarantees that the SSN phase initiated

---

[2]Algorithm 1 specifies this as PRM$^+$, but PRM$^+$ can be replaced by any first-order method. See Appendix B for further discussion and comparisons.

at that lifted point exhibits local quadratic convergence when applied to our residual operator. Thus, our hybrid method effectively inherits the best-of-both worlds: global progress from FOMs and rapid local convergence of the regularized SSN method.

## 4. Numerical Results

We conduct numerical experiments on a variety of matrix games, including Kuhn poker (Kuhn, 1950), three families of matrix games arising from layered graph security settings (Černý et al., 2024), and random normal and uniform matrix games. In the realistic game instances, such as Kuhn poker and layered security games, PRM$^+$ rapidly achieves high-precision last-iterate convergence, making them less suitable for evaluating the benefits of our hybrid SSN algorithm. Accordingly, in this section, we focus on random normal and uniform matrix games, where PRM$^+$ struggles to efficiently converge to high-precision Nash equilibria.

Our hybrid algorithm is evaluated under two main schemes: **PSSN v1** and **PSSN v2**. These schemes perform a one-time switch from PRM$^+$ (QA) to the regularized semi-smooth Newton method once the duality gap of the PRM$^+$ iterates drops under a fixed threshold, but differ in their updating of the damping parameter; the SSN stage of PSSN v1 is initialized with a fixed damping parameter, whereas the SSN stage of PSSN v2 uses a damping parameter tuned by the PRM$^+$ steps. For completeness, we also include a third scheme, **HPSSN**, in the tables below. HPSSN follows a similar warm-starting strategy but alternates back and forth between PRM$^+$ and SSN at different rates based on the duality gap, in an effort to reduce the number of costly DRSSN steps while still exploiting curvature information near equilibrium. Although this adaptive alternation is conceptually appealing, in practice it is difficult to tune effectively: varying switching thresholds and the number of DRSSN steps does not consistently yield strong results without extensive tuning.

Our algorithms are compared to PRM$^+$ last-iterate (LI) and quadratic averaging (QA) baselines, which each use alternation, and are run for $5 \times 10^5$ iterations. All algorithms are initialized at $(x, y) = ((1/n)\mathbf{1}_n, (1/m)\mathbf{1}_m)$.

For succinctness, we report numerical results averaged over 10 independently generated random uniform and normal matrix games, of size $400 \times 800$. These games are representative of the general trends we observe across different random matrix game sizes and highlight the efficacy of our hybrid approach over FOMs for achieving high-precision solutions. In particular, despite the one-time cost of calculating resolvents to initialize our DRSSN method, our PSSN algorithms achieve significant speedups in the high-accuracy regime ($10^{-8}$ duality gap or lower), underscoring

the practical benefit of incorporating curvature information. Note that, once DRSSN reaches a precision of around $10^{-8}$, it takes almost no additional time to reach higher levels of precision, e.g. $10^{-12}$. Thus, our numerical experiments corroborate the theoretical superlinear convergence guarantee of our methods.

*Table 1.* Averaged clock times (in seconds) for $400 \times 800$ random uniform matrix games

| Method | Time to Reach Duality Gap Tolerance | | | | | |
| | $10^{-2}$ | $10^{-4}$ | $10^{-6}$ | $10^{-8}$ | $10^{-10}$ | $10^{-12}$ |
|---|---|---|---|---|---|---|
| PRM$^+$ (LI) | 0.007 | 0.059 | 3.848 | 21.69 | | |
| PRM$^+$ (QA) | 0.006 | 0.030 | 0.293 | 6.424 | 21.97 | |
| PSSN v1 | 0.089 | 0.376 | 3.463 | 5.767 | 5.922 | 5.940 |
| PSSN v2 | 0.090 | 0.385 | 3.638 | 4.290 | 4.307 | 4.323 |
| HPSSN | 0.101 | 0.399 | 3.905 | 5.068 | 5.082 | 5.092 |

*Table 2.* Averaged clock times (in seconds) for $400 \times 800$ random normal matrix games

| Method | Time to Reach Duality Gap Tolerance | | | | | |
| | $10^{-2}$ | $10^{-4}$ | $10^{-6}$ | $10^{-8}$ | $10^{-10}$ | $10^{-12}$ |
|---|---|---|---|---|---|---|
| PRM$^+$ (LI) | 0.006 | 0.087 | 3.325 | 23.51 | | |
| PRM$^+$ (QA) | 0.006 | 0.036 | 0.366 | 4.923 | 19.83 | |
| PSSN v1 | 0.128 | 0.690 | 4.238 | 5.267 | 5.289 | 5.306 |
| PSSN v2 | 0.128 | 0.691 | 3.983 | 5.055 | 5.077 | 5.094 |
| HPSSN | 0.125 | 0.572 | 3.895 | 4.786 | 4.790 | 4.815 |

Our numerical experiments focus on comparisons with PRM$^+$ rather than LP-based approaches because LP solvers are not competitive in the regimes relevant to modern game solving, where the repeated solution of large linear systems becomes prohibitively expensive. Consequently, first-order regret-minimization methods such as PRM$^+$ and its variants have long served as the baseline in the game solving literature due to their scalability. Although LP methods can solve small games efficiently, it is standard to benchmark algorithms on instances that are small enough to support careful ablations, yet representative of methods intended to scale beyond what is achievable with black-box LP solvers; see, for example, Farina et al. (2019); Chakrabarti et al. (2024); Fang et al. (2025); Cai et al. (2025), among others. Our experiments follow this convention and assess performance relative to the state-of-the-art first-order method PRM$^+$. For context and completeness, we include comparisons to other first-order methods in the appendix, which also contains further numerical experiments with detailed descriptions of all algorithms and game instances.

# 5. Conclusion

We develop the first direct second-order method for game solving and formalize a rigorous framework for transitioning from global first-order progress to local superlinear convergence via a semi-smooth Newton (SSN) method. We show that progress made by any first-order method that reduces the duality gap can be systematically leveraged to warm-start the SSN phase, leading to fast local convergence and high-precision solutions. Our results represent a conceptual and technical advance beyond the existing game-solving literature, which has relied almost exclusively on first-order methods for nearly two decades.

We view scalability as an important direction for further development. The present work should be viewed as a foundational step: it demonstrates that second-order information can be effectively incorporated into game-solving pipelines and can substantially accelerate the computation of high-accuracy solutions. To further improve performance at larger scales, it may be necessary to refine the algorithmic framework — for example, by incorporating inexact linear system solves in place of exact solves, or by developing more refined heuristics for switching and adaptive damping parameter updates.

More broadly, we believe this work opens a promising direction for incorporating second-order information into scalable game-solving algorithms. An important next step is to extend our framework beyond matrix games to extensive-form games.

## Impact Statement

This paper presents work whose goal is to advance the field of Machine Learning. There are many potential societal consequences of our work, none which we feel must be specifically highlighted here.

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

## A. Proofs

### A.1. Proofs for Section 2

Recall that the DRS operator is given by

$$T_{\text{DRS}}^\gamma = \text{Id} - J_{N_S}^\gamma + J_F^\gamma \circ (2J_{N_S}^\gamma - \text{Id}),$$

and the residual operator is given by $R_{\text{DRS}}^\gamma = \text{Id} - T_{\text{DRS}}^\gamma$.

Because $F(\cdot)$ is linear and $N_S(\cdot)$ is a normal cone operator, their resolvents simplify to

$$J_F^\gamma(z) = \begin{bmatrix} I & \gamma A \\ -\gamma A^\top & I \end{bmatrix}^{-1} z, \quad J_{N_S}^\gamma(z) = \Pi_S(z).$$

As a consequence, the DRS operator takes the form

$$T_{\text{DRS}}^\gamma = \text{Id} - \Pi_S + \begin{bmatrix} I & \gamma A \\ -\gamma A^\top & I \end{bmatrix}^{-1} (2\Pi_S - \text{Id}).$$

*Proof of Lemma 2.4.* For any $z_1, z_2$, we have

$$\|R_{\text{DRS}}^\gamma(z_1) - R_{\text{DRS}}^\gamma(z_2)\|^2 = \|(z_1 - z_2) - (T_{\text{DRS}}^\gamma(z_1) - T_{\text{DRS}}^\gamma(z_2))\|^2$$
$$= \|z_1 - z_2\|^2 + \|T_{\text{DRS}}^\gamma(z_1) - T_{\text{DRS}}^\gamma(z_2)\|^2 - 2(z_1 - z_2)^\top (T_{\text{DRS}}^\gamma(z_1) - T_{\text{DRS}}^\gamma(z_2)).$$

It is known that $T_{\text{DRS}}^\gamma$ is firmly nonexpansive (Ryu & Yin, 2022). Equivalently,

$$\|T_{\text{DRS}}^\gamma(z_1) - T_{\text{DRS}}^\gamma(z_2)\|^2 \le (z_1 - z_2)^\top (T_{\text{DRS}}^\gamma(z_1) - T_{\text{DRS}}^\gamma(z_2)).$$

Putting these pieces together yields

$$\|R_{\text{DRS}}^\gamma(z_1) - R_{\text{DRS}}^\gamma(z_2)\|^2 \le \|z_1 - z_2\|^2 - (z_1 - z_2)^\top (T_{\text{DRS}}^\gamma(z_1) - T_{\text{DRS}}^\gamma(z_2))$$
$$= (z_1 - z_2)^\top (R_{\text{DRS}}^\gamma(z_1) - R_{\text{DRS}}^\gamma(z_2)).$$

This implies that $R_{\text{DRS}}^\gamma(\cdot)$ is monotone.

We also have

$$(z_1 - z_2)^\top (R_{\text{DRS}}^\gamma(z_1) - R_{\text{DRS}}^\gamma(z_2)) \le \|z_1 - z_2\|\|R_{\text{DRS}}^\gamma(z_1) - R_{\text{DRS}}^\gamma(z_2)\|.$$

Thus, we have

$$\|R_{\text{DRS}}^\gamma(z_1) - R_{\text{DRS}}^\gamma(z_2)\| \le \|z_1 - z_2\|.$$

This implies that $R_{\text{DRS}}^\gamma(\cdot)$ is 1-Lipschitz. □

*Proof of Lemma 2.7.* We first show that the residual operator $R_{\text{DRS}}^\gamma(\cdot)$ is a piecewise affine mapping. By construction, it is a linear transformation constructed from the linear resolvent $J_F^\gamma(\cdot)$ and $J_{N_S}^\gamma(\cdot) = \Pi_S(\cdot)$, the Euclidean projection operator onto the product of two simplices, which is piecewise affine (Rockafellar & Wets, 1998, Example 12.31).

The strong semi-smoothness of $R_{\text{DRS}}^\gamma(\cdot)$ now follows from Facchinei & Pang (2003, Vol. II, Proposition 7.4.7): since $R_{\text{DRS}}^\gamma(\cdot)$ is a piecewise affine mapping, it is strongly semi-smooth. □

### A.2. Proofs for Section 3

**Lemma A.1** (Equivalence between Projection and Normal Cone). *Let $C$ be a closed and convex set. Then $p \in C$ is the projection of $x$ onto $C$ if and only if $x - p \in N_C(p)$.*

*Proof.* It is well-known that $p \in C$ is the projection of $x$ onto $C$ if and only if

$$\langle y - p, x - p \rangle \le 0$$

for all $y \in C$. By the definition of the normal cone, this is equivalent to the condition that $x - p \in N_C(p)$. □

*Proof of Theorem 3.1 (i).* From Equation (3), a fixed point $z$ of $T_{\text{DRS}}^\gamma$ satisfies

$$z = z - \Pi_{\mathcal{S}}(z) + (I + \gamma F)^{-1}(2\Pi_{\mathcal{S}}(z) - z)$$

or equivalently, $(I + \gamma F)^{-1}(2\Pi_{\mathcal{S}}(z) - z) = \Pi_{\mathcal{S}}(z)$. It follows that

$$(I + \gamma F)(\Pi_{\mathcal{S}}(z)) = 2\Pi_{\mathcal{S}}(z) - z.$$

Expanding and simplifying, we get that $\Pi_{\mathcal{S}}(z) + \gamma F(\Pi_{\mathcal{S}}(z)) = 2\Pi_{\mathcal{S}}(z) - z$, or equivalently,

$$\frac{1}{\gamma}(\Pi_{\mathcal{S}}(z) - z) = F(\Pi_{\mathcal{S}}(z)). \tag{9}$$

Furthermore, note that by the first-order optimality condition for projections, we have

$$\frac{1}{\gamma}\langle z - \Pi_{\mathcal{S}}(z), \tilde{z} - \Pi_{\mathcal{S}}(z)\rangle \leq 0$$

for all $\tilde{z} \in \mathcal{S}$, or equivalently,

$$\frac{1}{\gamma}(z - \Pi_{\mathcal{S}}(z)) \in N_{\mathcal{S}}(\Pi_{\mathcal{S}}(z)). \tag{10}$$

Summing Equation (9) and Equation (10), we get the desired result,

$$0 \in F(\Pi_{\mathcal{S}}(z)) + N_{\mathcal{S}}(\Pi_{\mathcal{S}}(z)).$$

By the optimality condition for a Nash equilibrium, $\hat{z} = \Pi_{\mathcal{S}}(z)$ is a Nash equilibrium. □

*Proof of Theorem 3.1 (ii).* Since $\hat{z}$ is a Nash equilibrium, we have that $0 \in F(\hat{z}) + N_{\mathcal{S}}(\hat{z})$. Equivalently, $u := -F(\hat{z}) \in N_{\mathcal{S}}(\hat{z})$.

Consider $z := \hat{z} - \gamma F(\hat{z}) = \hat{z} + \gamma u$. Since $\hat{z} \in \mathcal{S}$ and $u \in N_{\mathcal{S}}(\hat{z})$ implies that $(\hat{z} + \gamma u) - \hat{z} = \gamma u \in N_{\mathcal{S}}(\hat{z})$, by Lemma A.1, we have that $\hat{z} = \Pi_{\mathcal{S}}(\hat{z} + \gamma u)$. It follows that $\Pi_{\mathcal{S}}(z) = \Pi_{\mathcal{S}}(\hat{z} + \gamma u) = \hat{z}$, and we have that

$$
\begin{aligned}
2\Pi_{\mathcal{S}}(z) - z &= 2\Pi_{\mathcal{S}}(\hat{z} + \gamma u) - (\hat{z} + \gamma u) \\
&= 2\hat{z} - (\hat{z} + \gamma u) \\
&= \hat{z} - \gamma u.
\end{aligned}
$$

Applying $T_{\text{DRS}}^\gamma$ to the lifted point $z$, we get that

$$
\begin{aligned}
T_{\text{DRS}}^\gamma(z) &= z - \Pi_{\mathcal{S}}(z) + (I + \gamma F)^{-1}(2\Pi_{\mathcal{S}}(z) - z) \\
&= (\hat{z} + \gamma u) - \hat{z} + (I + \gamma F)^{-1}(\hat{z} - \gamma u) \\
&= (\hat{z} + \gamma u) - \hat{z} + (I + \gamma F)^{-1}(\hat{z} + \gamma F(\hat{z})) \\
&= (\hat{z} + \gamma u) - \hat{z} + \hat{z} \\
&= \hat{z} + \gamma u = z
\end{aligned}
$$

where the third line follows from the definition $u = -F(\hat{z})$. Thus, $z = \hat{z} - \gamma F(\hat{z})$ is a fixed point of $T_{\text{DRS}}^\gamma$. □

*Proof of Theorem 3.3.* For any $i \in \mathcal{A}$, the closed-form simplex projection formula gives

$$x_i = p_i - \alpha, \text{ where } \alpha = \frac{1}{|\mathcal{A}|}\left(\sum_{j \in \mathcal{A}} p_j - 1\right)$$

For $i \notin \mathcal{A}$, we have $x_i = 0$. Differentiating with respect to $p_j$, we get, for $i \in \mathcal{A}$,

$$G_{ij} = \frac{\partial x_i}{\partial p_j} = \begin{cases} 1 - \frac{1}{|\mathcal{A}|}, & \text{if } i = j \in \mathcal{A} \\ -\frac{1}{|\mathcal{A}|} & \text{if } j \in \mathcal{A}, \ i \neq j \\ 0 & \text{if } j \notin \mathcal{A} \end{cases}$$

On the other hand, for $i \notin \mathcal{A}$, we trivially have $G_{ij} = 0$ for all $j$. Equivalently, the Jacobian admits the compact representation

$$G = \mathrm{diag}(a) - \frac{1}{\|a\|_1} a a^\top$$

where $a$ is the indicator vector of the active set $\mathcal{A}$. $\qquad\square$

**Lemma A.2.** *For any two points $z_1 = (x_1 \, y_1) \in \mathcal{S}$ and $z_2 = (x_2 \, y_2) \in \mathcal{S}$, we have*

$$|\mathrm{gap}(z_1) - \mathrm{gap}(z_2)| \le \sqrt{2}\|A\| \cdot \|z_1 - z_2\|.$$

*Proof of Lemma A.2.* By definition,

$$\mathrm{gap}(x, y) = \max_{\bar{y} \in Y} x^\top A \bar{y} - \min_{\bar{x} \in X} \bar{x}^\top A y.$$

Let us define $\Phi(x) := \max_{\bar{y} \in Y} x^\top A \bar{y}$ and $\Psi(y) := \min_{\bar{x} \in X} \bar{x}^\top A y$, so that $\mathrm{gap}(x, y) = \Phi(x) - \Psi(y)$.

We have that

$$
\begin{aligned}
\Phi(x_1) - \Phi(x_2) &= \max_{\bar{y} \in Y} x_1^\top A \bar{y} - \max_{\hat{y} \in Y} x_2^\top A \hat{y} \\
&\le \max_{\bar{y} \in Y} \left[ x_1^\top A \bar{y} - x_2^\top A \bar{y} \right] \\
&= \max_{\bar{y} \in Y} \langle x_1 - x_2, A\bar{y} \rangle \\
&\le \|A\| \cdot \|x_1 - x_2\|.
\end{aligned}
$$

where the last line follows by the Cauchy-Schwarz inequality and the fact that $\|y\| \le 1$ for all $y \in \Delta^m$. The above result also implies that

$$\Phi(x_2) - \Phi(x_1) \le \|A\| \cdot \|x_2 - x_1\|.$$

Combining the above two results, we get that

$$|\Phi(x_1) - \Phi(x_2)| \le \|A\| \cdot \|x_1 - x_2\|. \tag{11}$$

Similarly, we have that

$$
\begin{aligned}
\Psi(y_1) - \Psi(y_2) &= \min_{\bar{x} \in X} \bar{x}^\top A y_1 - \min_{\hat{x} \in X} \hat{x}^\top A y_2 \\
&\le \max_{\bar{x} \in X} \left[ \bar{x}^\top A y_1 - \bar{x}^\top A y_2 \right] \\
&= \max_{\bar{x} \in X} \langle \bar{x}, A(y_1 - y_2) \rangle \\
&\le \|A\| \cdot \|y_1 - y_2\|
\end{aligned}
$$

where the last line follows by the Cauchy-Schwarz inequality and the fact that $\|x\| \le 1$ for all $x \in \Delta^n$. The above result also implies that

$$\Psi(y_2) - \Psi(y_1) \le \|A\| \cdot \|y_2 - y_1\|.$$

Combining the above two results, we get that

$$|\Psi(y_1) - \Psi(y_2)| \le \|A\| \cdot \|y_1 - y_2\|. \tag{12}$$

Since

$$
\begin{aligned}
\mathrm{gap}(z_1) - \mathrm{gap}(z_2) &= \mathrm{gap}(x_1, y_1) - \mathrm{gap}(x_2, y_2) \\
&= (\Phi(x_1) - \Psi(y_1)) - (\Phi(x_2) - \Psi(y_2)) \\
&= (\Phi(x_1) - \Phi(x_2)) - (\Psi(y_1) - \Psi(y_2)),
\end{aligned}
$$

we have that

$$
\begin{aligned}
|\mathrm{gap}(z_1) - \mathrm{gap}(z_2)| &= |\Phi(x_1) - \Phi(x_2) + \Psi(y_1) - \Psi(y_2)| \\
&\leq |\Phi(x_1) - \Phi(x_2)| + |\Psi(y_1) - \Psi(y_2)| \\
&\leq \|A\| \cdot (\|x_1 - x_2\| + \|y_1 - y_2\|) \\
&\leq \sqrt{2}\|A\| \cdot \|z_1 - z_2\|
\end{aligned}
$$

where the third line follows from applying the bounds of Equation (11) and Equation (12) and the final line follows from an application of the inequality $a + b \leq \sqrt{2}\sqrt{a^2 + b^2}$ where $a = \|x_1 - x_2\|$, $b = \|y_1 - y_2\|$, and so $\sqrt{a^2 + b^2} = \|z_1 - z_2\|$. $\quad\square$

*Proof of Theorem 3.6 (i).* Let $z^*$ be the closest optimal solution to $z$ in $Z^*$. Rewriting the given condition $\|R_{\mathrm{DRS}}^\gamma(z)\| \geq \tau\|z - z^*\|$, we have

$$
\|z - z^*\| \leq \frac{\|R_{\mathrm{DRS}}^\gamma(z)\|}{\tau}. \tag{13}
$$

Let us define $\hat{z} := \Pi_{\mathcal{S}}(z)$ and $\hat{z}^* := \Pi_{\mathcal{S}}(z^*)$. Since projection is a nonexpansive operator, we have that

$$
\|\hat{z} - \hat{z}^*\| \leq \|z - z^*\|. \tag{14}
$$

Furthermore, by Lemma A.2 applied to $\hat{z}$ and $\hat{z}^*$, we have that

$$
|\mathrm{gap}(\hat{z}) - \mathrm{gap}(\hat{z}^*)| \leq \sqrt{2}\|A\| \cdot \|\hat{z} - \hat{z}^*\| \tag{15}
$$

Combining the results of Equation (13), Equation (14), and Equation (15), we get that

$$
|\mathrm{gap}(\hat{z}) - \mathrm{gap}(\hat{z}^*)| \leq \sqrt{2}\|A\| \cdot \|\hat{z} - \hat{z}^*\| \leq \sqrt{2}\|A\| \cdot \|z - z^*\| \leq \frac{\sqrt{2}\|A\| \cdot \|R_{\mathrm{DRS}}^\gamma(z)\|}{\tau}.
$$

By Theorem 3.1 (i), $\mathrm{gap}(\hat{z}^*) = 0$. Thus, we conclude that

$$
|\mathrm{gap}(\hat{z})| = |\mathrm{gap}(\Pi_{\mathcal{S}}(z))| \leq \frac{\sqrt{2}\|A\| \cdot \|R_{\mathrm{DRS}}^\gamma(z)\|}{\tau}. \qquad\square
$$

*Proof of Theorem 3.6 (ii).* Let $\hat{z}^*$ be the closest Nash equilibrium to $\hat{z}$, so that $\mathrm{dist}(\hat{z}, \hat{Z}^*) = \|\hat{z} - \hat{z}^*\|$. By Equation (7), we have that

$$
\|\hat{z} - \hat{z}^*\| \leq \frac{\mathrm{gap}(\hat{z})}{\delta(A)}. \tag{16}
$$

Consider the lifted points $z := \hat{z} - \gamma F(\hat{z})$ and $z^* := \hat{z}^* - \gamma F(\hat{z}^*)$ of $\hat{z}$ and $\hat{z}^*$, respectively. Note that

$$
\begin{aligned}
\|z - z^*\| &= \|(\hat{z} - \gamma F(\hat{z})) - (\hat{z}^* - \gamma F(\hat{z}^*))\| \\
&= \|\hat{z} - \hat{z}^* - \gamma(F(\hat{z}) - F(\hat{z}^*))\| \\
&\leq \|\hat{z} - \hat{z}^*\| + \gamma\|A\| \cdot \|\hat{z} - \hat{z}^*\| \\
&= (1 + \gamma\|A\|)\|\hat{z} - \hat{z}^*\|
\end{aligned} \tag{17}
$$

where the third line follows from the triangle inequality and the definition of the operator $F$.

Finally, by Lemma 2.4 (ii), we know that $R_{\mathrm{DRS}}^\gamma(\cdot)$ is 1-Lipschitz, so

$$
\|R_{\mathrm{DRS}}^\gamma(z)\| = \|R_{\mathrm{DRS}}^\gamma(z) - R_{\mathrm{DRS}}^\gamma(z^*)\| \leq \|z - z^*\|. \tag{18}
$$

where the first equality follows by Theorem 3.1 (ii): $\hat{z}^*$ is a Nash equilibrium so its lift $z^*$ satisfies $R_{\mathrm{DRS}}^\gamma(z^*) = 0$.

Thus, we have that

$$
\begin{aligned}
\|R_{\mathrm{DRS}}^\gamma(z)\| &\leq \|z - z^*\| \\
&\leq (1 + \gamma\|A\|)\|\hat{z} - \hat{z}^*\| \\
&\leq \frac{(1 + \gamma\|A\|)\mathrm{gap}(\hat{z})}{\delta(A)}
\end{aligned}
$$

where the first line follows from Equation (18), the second line follows from Equation (17), and the final line follows from Equation (16). $\quad\square$

*Proof of Theorem 3.11.* Let $z_k^*$ be the closest optimal solution (zero of $R_{\mathrm{DRS}}^\gamma$) to $z_k$. Note that by definition, for $G_k \in \partial R_{\mathrm{DRS}}^\gamma(z_k)$, we have

$$\Delta z_k = -(G_k + \theta \|R_{\mathrm{DRS}}^\gamma(z_k)\| I)^{-1} R_{\mathrm{DRS}}^\gamma(z_k).$$

Adding and subtracting $z_k^* - z_k$ and factoring, we get that

$$
\begin{aligned}
\Delta z_k &= -(G_k + \theta \|R_{\mathrm{DRS}}^\gamma(z_k)\| I)^{-1} R_{\mathrm{DRS}}^\gamma(z_k) \\
&= -(G_k + \theta \|R_{\mathrm{DRS}}^\gamma(z_k)\| I)^{-1} R_{\mathrm{DRS}}^\gamma(z_k) - (z_k^* - z_k) + (z_k^* - z_k) \\
&= -(G_k + \theta \|R_{\mathrm{DRS}}^\gamma(z_k)\| I)^{-1} \left[ R_{\mathrm{DRS}}^\gamma(z_k) + (G_k + \theta \|R_{\mathrm{DRS}}^\gamma(z_k)\| I)(z_k^* - z_k) \right] + (z_k^* - z_k).
\end{aligned}
$$

Taking the norm of both sides, it follows that

$$\|\Delta z_k\| = \left\| -(G_k + \theta \|R_{\mathrm{DRS}}^\gamma(z_k)\| I)^{-1} \left[ R_{\mathrm{DRS}}^\gamma(z_k) + (G_k + \theta \|R_{\mathrm{DRS}}^\gamma(z_k)\| I)(z_k^* - z_k) \right] + (z_k^* - z_k) \right\|.$$

Applying the Triangle Inequality to the right-hand side and expanding gives

$$
\begin{aligned}
\|\Delta z_k\| &= \left\| -(G_k + \theta \|R_{\mathrm{DRS}}^\gamma(z_k)\| I)^{-1} \left[ R_{\mathrm{DRS}}^\gamma(z_k) + (G_k + \theta \|R_{\mathrm{DRS}}^\gamma(z_k)\| I)(z_k^* - z_k) \right] + (z_k^* - z_k) \right\| \\
&\leq \|(G_k + \theta \|R_{\mathrm{DRS}}^\gamma(z_k)\| I)^{-1}\| \cdot \|R_{\mathrm{DRS}}^\gamma(z_k) + G_k(z_k^* - z_k) + \theta \|R_{\mathrm{DRS}}^\gamma(z_k)\|(z_k^* - z_k)\| \\
&\quad + \|z_k^* - z_k\|.
\end{aligned}
$$

Applying Lemma 3.7 on the first factor, we get that

$$
\begin{aligned}
\|\Delta z_k\| &\leq \|(G_k + \theta \|R_{\mathrm{DRS}}^\gamma(z_k)\| I)^{-1}\| \cdot \|R_{\mathrm{DRS}}^\gamma(z_k) + G_k(z_k^* - z_k) + \theta \|R_{\mathrm{DRS}}^\gamma(z_k)\|(z_k^* - z_k)\| \\
&\quad + \|z_k^* - z_k\| \\
&\leq \frac{1}{\theta \|R_{\mathrm{DRS}}^\gamma(z_k)\|} \|R_{\mathrm{DRS}}^\gamma(z_k) + G_k(z_k^* - z_k) + \theta \|R_{\mathrm{DRS}}^\gamma(z_k)\|(z_k^* - z_k)\| + \|z_k^* - z_k\| \\
&\leq \frac{1}{\theta \|R_{\mathrm{DRS}}^\gamma(z_k)\|} \left( \|R_{\mathrm{DRS}}^\gamma(z_k) - G_k(z_k - z_k^*)\| + \theta \|R_{\mathrm{DRS}}^\gamma(z_k)\| \cdot \|z_k - z_k^*\| \right) + \|z_k^* - z_k\|.
\end{aligned}
$$

where the final line follows from the Triangle Inequality. From Lemma 3.8, we have that

$$\|R_{\mathrm{DRS}}^\gamma(z_k) - G_k(z_k - z_k^*)\| \leq L_2 \|z_k - z_k^*\|^2$$

for some $L_2 > 0$. Plugging this result in above, we get that

$$
\begin{aligned}
\|\Delta z_k\| &\leq \frac{1}{\theta \|R_{\mathrm{DRS}}^\gamma(z_k)\|} \left( \|R_{\mathrm{DRS}}^\gamma(z_k) - G_k(z_k - z_k^*)\| + \theta \|R_{\mathrm{DRS}}^\gamma(z_k)\| \cdot \|z_k - z_k^*\| \right) + \|z_k^* - z_k\| \\
&\leq \frac{1}{\theta \|R_{\mathrm{DRS}}^\gamma(z_k)\|} \left( L_2 \|z_k - z_k^*\|^2 + \theta \|R_{\mathrm{DRS}}^\gamma(z_k)\| \cdot \|z_k - z_k^*\| \right) + \|z_k^* - z_k\| \qquad (19) \\
&= 2\|z_k - z_k^*\| + \frac{L_2}{\theta \|R_{\mathrm{DRS}}^\gamma(z_k)\|} \|z_k - z_k^*\|^2.
\end{aligned}
$$

Since $R_{\mathrm{DRS}}^\gamma(\cdot)$ satisfies the local error-bound (Lemma 3.10), we have that

$$\|R_{\mathrm{DRS}}^\gamma(z_k)\| = \|R_{\mathrm{DRS}}^\gamma(z_k) - R_{\mathrm{DRS}}^\gamma(z_k^*)\| \geq \tau \|z_k - z_k^*\|.$$

Taking the reciprocal of both sides gives

$$\frac{1}{\|R_{\mathrm{DRS}}^\gamma(z_k)\|} \leq \frac{1}{\tau \|z_k - z_k^*\|}.$$

Plugging this bound into the result of Equation (19) gives

$$
\begin{aligned}
\|\Delta z_k\| &\leq 2\|z_k - z_k^*\| + \frac{L_2}{\theta \|R_{\mathrm{DRS}}^\gamma(z_k)\|} \|z_k - z_k^*\|^2 \\
&\leq 2\|z_k - z_k^*\| + \frac{L_2}{\theta \tau \|z_k - z_k^*\|} \|z_k - z_k^*\|^2 \qquad (20) \\
&= \left( 2 + \frac{L_2}{\theta \tau} \right) \|z_k - z_k^*\|.
\end{aligned}
$$

We now focus on the norm of the residual map at consecutive iterates $z_k$ and $z_{k+1}$. By definition,

$$\|R_{\text{DRS}}^\gamma(z_{k+1})\| = \|R_{\text{DRS}}^\gamma(z_k + \Delta z_k)\|.$$

Adding and subtracting $R_{\text{DRS}}^\gamma(z_k) + G_k \Delta z_k$ and then applying the Triangle Inequality, we get that

$$
\begin{aligned}
\|R_{\text{DRS}}^\gamma(z_{k+1})\| &= \|R_{\text{DRS}}^\gamma(z_k + \Delta z_k)\| \\
&= \|R_{\text{DRS}}^\gamma(z_k + \Delta z_k) - R_{\text{DRS}}^\gamma(z_k) - G_k \Delta z_k + R_{\text{DRS}}^\gamma(z_k) + G_k \Delta z_k\| \\
&\leq \|R_{\text{DRS}}^\gamma(z_k + \Delta z_k) - R_{\text{DRS}}^\gamma(z_k) - G_k \Delta z_k\| + \|R_{\text{DRS}}^\gamma(z_k) + G_k \Delta z_k\|.
\end{aligned}
\tag{21}
$$

Since $R_{\text{DRS}}^\gamma(\cdot)$ satisfies Lemma 3.9, we have

$$\|R_{\text{DRS}}^\gamma(z_k + \Delta z_k) - R_{\text{DRS}}^\gamma(z_k) - G_k \Delta z_k\| \leq L_3 \|\Delta z_k\|^2. \tag{22}$$

On the other hand, by the definition of $\Delta z_k$, we have that

$$R_{\text{DRS}}^\gamma(z_k) + (G_k + \theta \|R_{\text{DRS}}^\gamma(z_k)\| I)\Delta z_k = 0.$$

Equivalently, after expanding, rearranging, and taking the norm of both sides, we have that

$$\|R_{\text{DRS}}^\gamma(z_k) + G_k \Delta z_k\| = \theta \|R_{\text{DRS}}^\gamma(z_k)\| \cdot \|\Delta z_k\|. \tag{23}$$

Plugging in the results of Equation (22) and Equation (23) into the final line of Equation (21), we get that

$$
\begin{aligned}
\|R_{\text{DRS}}^\gamma(z_{k+1})\| &\leq \|R_{\text{DRS}}^\gamma(z_k + \Delta z_k) - R_{\text{DRS}}^\gamma(z_k) - G_k \Delta z_k\| + \|R_{\text{DRS}}^\gamma(z_k) + G_k \Delta z_k\| \\
&\leq L_3 \|\Delta z_k\|^2 + \theta \|R_{\text{DRS}}^\gamma(z_k)\| \cdot \|\Delta z_k\|.
\end{aligned}
\tag{24}
$$

Substituting the result from Equation (20) into the result from Equation (24) and factoring, we get that

$$
\begin{aligned}
\|R_{\text{DRS}}^\gamma(z_{k+1})\| &\leq L_3 \|\Delta z_k\|^2 + \theta \|R_{\text{DRS}}^\gamma(z_k)\| \cdot \|\Delta z_k\| \\
&\leq L_3 \left(2 + \frac{L_2}{\theta\tau}\right)^2 \|z_k - z_k^*\|^2 + \theta \|R_{\text{DRS}}^\gamma(z_k)\| \left(2 + \frac{L_2}{\theta\tau}\right) \|z_k - z_k^*\| \\
&\leq \left[L_3 \left(2 + \frac{L_2}{\theta\tau}\right)^2 + \theta L_1 \left(2 + \frac{L_2}{\theta\tau}\right)\right] \|z_k - z_k^*\|^2
\end{aligned}
\tag{25}
$$

where the final line follows as $R_{\text{DRS}}^\gamma(\cdot)$ is 1-Lipschitz (Lemma 2.4 (ii)), or equivalently,

$$\|R_{\text{DRS}}^\gamma(z_k)\| = \|R_{\text{DRS}}^\gamma(z_k) - R_{\text{DRS}}^\gamma(z_k^*)\| \leq L_1 \|z_k - z_k^*\|.$$

Since $R_{\text{DRS}}^\gamma(\cdot)$ satisfies the local error-bound condition (Lemma 3.10), we have that

$$\|R_{\text{DRS}}^\gamma(z_k)\| = \|R_{\text{DRS}}^\gamma(z_k) - R_{\text{DRS}}^\gamma(z_k^*)\| \geq \tau \|z_k - z_k^*\|.$$

Squaring both sides and rearranging, it follows that

$$\|z_k - z_k^*\|^2 \leq \frac{\|R_{\text{DRS}}^\gamma(z_k)\|^2}{\tau^2}$$

Using the above result to bound the right-hand side of Equation (25) above, we get that

$$
\begin{aligned}
\|R_{\text{DRS}}^\gamma(z_{k+1})\| &\leq \left[L_3 \left(2 + \frac{L_2}{\theta\tau}\right)^2 + \theta L_1 \left(2 + \frac{L_2}{\theta\tau}\right)\right] \|z_k - z_k^*\|^2 \\
&\leq \frac{1}{\tau^2}\left[L_3 \left(2 + \frac{L_2}{\theta\tau}\right)^2 + \theta L_1 \left(2 + \frac{L_2}{\theta\tau}\right)\right] \|R_{\text{DRS}}^\gamma(z_k)\|^2.
\end{aligned}
$$

which gives us the desired local quadratic convergence of Equation (8). $\qquad\square$

# B. Our Algorithm

## B.1. Predictive Regret Matching$^+$

We first provide an overview of the Predictive Regret Matching$^+$ (PRM$^+$) algorithm, and then discuss its effectiveness over other first-order methods for game solving.

**Description of PRM$^+$.** At a high level, the classical Regret Matching (RM) algorithm maintains a cumulative regret vector over actions, measuring how much better each pure action would have performed relative to the mixed strategy that is played. At each iteration, the algorithm assigns probability mass only to actions with positive cumulative regret, proportional to the magnitude of that regret. Regret Matching$^+$ (RM$^+$) is a simple yet powerful modification of this idea; after each update, the cumulative regrets are clipped at zero (using the operator $[\cdot]^+ = \max\{\cdot, 0\}$), thus preventing negative regrets from accumulating and yielding significantly improved empirical performance. The predictive variant PRM$^+$ further incorporates a prediction vector $m^t$ directly in its update. This allows the algorithm to estimate the next loss and adjust its iterate in anticipation, rather than reacting only after observing the realized loss $\ell^t$. As a result, PRM$^+$ exhibits faster convergence in practice.

Below, we provide the pseudocode of PRM$^+$, taken directly from Farina et al. (2021).

---

**Algorithm 3** (Predictive) regret matching$^+$ [(P)RM$^+$]

---

1: $z^0 \leftarrow 0 \in \mathbb{R}^n, \quad x^0 \leftarrow \frac{1}{n}\mathbf{1} \in \Delta^n$
2: **function** NextStrategy($m^t$)
3:     $\theta^t \leftarrow \left[z^{t-1} + \langle m^t, x^{t-1}\rangle\mathbf{1} - m^t\right]^+$                                     $\triangleright$ Set $m^t = 0$ for the non-predictive version
4:     **if** $\theta^t \neq 0$ **then**
5:         **return** $x^t \leftarrow \theta^t / \|\theta^t\|_1$
6:     **else**
7:         **return** $x^t \leftarrow$ arbitrary point in $\Delta^n$
8:     **end if**
9: **end function**
10: **function** ObserveLoss($\ell^t$)
11:     $z^t \leftarrow \left[z^{t-1} + \langle \ell^t, x^t\rangle\mathbf{1} - \ell^t\right]^+$
12: **end function**

---

The game solving literature also uses a variety of heuristics for stronger numerical performance. The *quadratic averaging* scheme uses a weighted average of past iterates in which the weight assigned to the iterate at time step $t$ is $t^2$, while the *last-iterate* scheme simply uses the last iterate without any averaging. Finally, *alternation* (or *alternating updates*) refers to updating the players sequentially within each iteration, which is commonly used to further improve convergence behavior in practice.

**Choice of PRM$^+$ as First-Order Method.** Our hybrid SSN framework is not tied to PRM$^+$ specifically. The theoretical conditions (Theorem 3.6) governing the transition to the second-order phase depend only on the quality of the current iterate, as measured by the duality gap. Consequently, any first-order method that reliably reduces the duality gap, including those guaranteeing last-iterate (or average) convergence in these zero-sum games, is compatible with our setup.

Our focus on PRM$^+$ in our experiments is motivated by the fact that it is the state-of-the-art practical first-order method for large-scale zero-sum games. In particular, the game-solving literature overwhelmingly finds that PRM$^+$ and its extensive-form variant PCFR$^+$ outperform other first-order methods on saddle-point problems. This is consistent with our own empirical evaluations. For completeness, we report the relative performances of extragradient (EG) and optimistic gradient descent-ascent (OGDA) against PRM$^+$ for the $400 \times 800$ random matrix games tested in the main body below.

In these experiments, we run both EG and OGDA for $10^5$ iterations; this choice is deliberate, as each iteration is more expensive than a PRM$^+$ iteration, and the differences in both runtime and performance are already clearly visible.

Across the ten random seeds tested, the symbol ($\dagger$) indicates that one seed failed to reach the specified duality-gap threshold, while the symbol ($*$) indicates that nine seeds failed to reach the threshold. EG and OGDA are substantially slower across the full range of tolerances, whereas PRM$^+$ reaches comparable accuracy orders of magnitude faster. These results further

*Table 3.* Averaged clock times (seconds) for $400 \times 800$ random matrix games.

*Random Normal*

| Method | Time to Reach Duality Gap Tolerance | | | | | |
|--------|--------|--------|--------|--------------|----------------|--------|
| | $10^{-2}$ | $10^{-4}$ | $10^{-5}$ | $9 \cdot 10^{-6}$ | $7.5 \cdot 10^{-6}$ | $10^{-6}$ |
| EG | 0.322 | 7.255 | 46.532 | 51.580 | 59.049 | $(*)$ |
| OGDA | 0.070 | 1.584 | 10.378 | 11.381 | 13.342 | $(*)$ |
| PRM$^+$ (LI) | 0.006 | 0.087 | 0.465 | 0.505 | 0.592 | 0.851 |
| PRM$^+$ (QA) | 0.006 | 0.036 | 0.110 | 0.116 | 0.131 | 0.165 |

*Random Uniform*

| Method | Time to Reach Duality Gap Tolerance | | | | |
|--------|--------|--------|--------|--------------|--------|
| | $10^{-2}$ | $10^{-4}$ | $10^{-5}$ | $9 \cdot 10^{-6}$ | $10^{-6}$ |
| EG | 0.436 | 9.645 | 65.302 | $(\dagger)$ | $(*)$ |
| OGDA | 0.099 | 2.140 | 14.361 | 16.340 | $(*)$ |
| PRM$^+$ (LI) | 0.007 | 0.059 | 0.301 | 0.344 | 3.848 |
| PRM$^+$ (QA) | 0.006 | 0.030 | 0.084 | 0.086 | 0.293 |

justify our use of PRM$^+$ (with alternation under both quadratic-averaging and last-iterate schemes) as the primary baseline in Table 1 and Table 2.

In summary, the strong empirical performance of PRM$^+$, together with its scalability to extensive-form games via PCFR$^+$, makes it a natural and representative choice for both benchmarking and warm-starting our hybrid algorithms. We provide further details on the integration of PRM$^+$ with our methods in the following sections.

## B.2. Adaptive Scheme

The $\beta_0$ coefficient serves as a contraction factor for the damping parameter $\lambda$, and its magnitude reflects the relative weight placed on second-order information from $G_k \in \partial R_{\mathrm{DRS}}^{\gamma}(x_k, y_k)$. As the iterates approach the optimal solution, the residual norm $\|R_{\mathrm{DRS}}^{\gamma}(x_k, y_k)\|$ decreases, in turn causing $\beta_0$ to shrink.

$\psi$ measures the quality of the Newton direction $\begin{bmatrix} \Delta x \\ \Delta y \end{bmatrix}$. A small value of $\psi$ indicates that the direction is not informative, while a large value indicates that the direction is informative and effectively reduces the residual norm. Based on the value of $\psi$, the adaptive scheme, mirroring that of Lin et al. (2024), for updating $\lambda_{k+1}$ is as follows:

$$\lambda_{k+1} = \begin{cases} \max\{\underline{\lambda}, \beta_0 \lambda_k\} & \text{if } \psi \geq \alpha_2 \\ \beta_1 \lambda_k & \text{if } \alpha_1 \leq \psi < \alpha_2 \\ \min\{\overline{\lambda}, \beta_2 \lambda_k\} & \text{if } \psi < \alpha_1 \end{cases}$$

Note that this adaptive update rule distinguishes three regimes:

- **High-quality direction** ($\psi \geq \alpha_2$): the step produces significant progress toward an optimal solution, so the damping parameter is moderately decreased by multiplying by a contraction factor $\beta_0 < 1$.

- **Moderate-quality direction** ($\alpha_1 \leq \psi < \alpha_2$): the step produces moderate but not significant progress towards an optimal solution, so the damping parameter is moderately increased by a factor $\beta_1 > 1$.

- **Low-quality direction** ($\psi < \alpha_1$): the step produces little to no progress toward an optimal solution, so the damping parameter is significantly increased by a factor $\beta_2 \gg 1$.

In our experiments, we use the parameters $\alpha_1 = 10^{-2}$, $\alpha_2 = 5$, $\beta_0 = \sqrt{\|R_{\mathrm{DRS}}^{\gamma}(x_k, y_k)\|}$, $\beta_1 = 2$, $\beta_2 = 5$, $\underline{\lambda} = 10^{-15}$, and $\overline{\lambda} = 10^{15}$.

## B.3. Description of Hybrid Algorithms

**PSSN v1** performs a one-time switch from PRM$^+$ with quadratic averaging and alternation to the DRSSN method once the duality gap drops under some specified fixed threshold. After switching, the SSN method is initialized with a fixed damping parameter of 1 and is run exclusively thereafter. This makes PSSN v1 the simplest hybrid variant, but its rigidity in fixing the damping parameter can limit its speed across different problem instances.

**PSSN v2** also performs a one-time switch from PRM$^+$ with quadratic averaging and alternation to the DRSSN method once the duality gap drops under some specified fixed threshold. Every 500 PRM$^+$ iterations, the damping parameter is iteratively updated using the adaptive scheme. After switching, the SSN method is initialized with this updated damping parameter. As a result, PSSN v2 is often more adapted to the dynamics of the problem, and often achieves better stability and efficiency than PSSN v1, although at the cost of some additional updating during the PRM$^+$ phase.

**HPSSN** adopts a more dynamic approach, alternating between PRM$^+$ and DRSSN throughout the run. At different stages, the algorithm attempts to lift a PRM$^+$ iterate and switch into DRSSN, taking only a few Newton steps before deciding whether superlinear convergence is being achieved. If not, it projects the current iterate back onto $\mathcal{S}$ and switches back to PRM$^+$. This strategy attempts to reduce the number of system solves from the SSN steps while still exploiting curvature information near equilibrium, striking a balance between the low per-iteration cost of first-order methods and the fast local convergence of second-order methods. We reiterate the discussion of Section 4: though this idea is conceptually appealing, it is difficult to tune effectively in practice and depends heavily on each problem instance. Given these shortcomings, we only include it in the $400 \times 800$ matrix game experiments.

# C. Additional Numerical Experiments

We provide both detailed descriptions for each game instance and additional tests comparing our algorithms to the PRM$^+$ baselines.

## C.1. Realistic Game Instances

**Kuhn poker** (Kuhn, 1950). At the beginning of the game, the two players each pay one chip to the pot, and are dealt a single private card from a deck containing three cards: Jack, Queen, and King. The first player can check or bet, putting an additional chip in the pot. Then, the second player can check or bet after the first player's check, or fold/call the first player's bet. If the second player bets, the first player can decide whether to fold or to call the bet. At the showdown, the player with the highest card who has not folded wins all the chips in the pot. The payoff matrix for Kuhn poker has dimension $27 \times 64$ with 690 nonzeros.

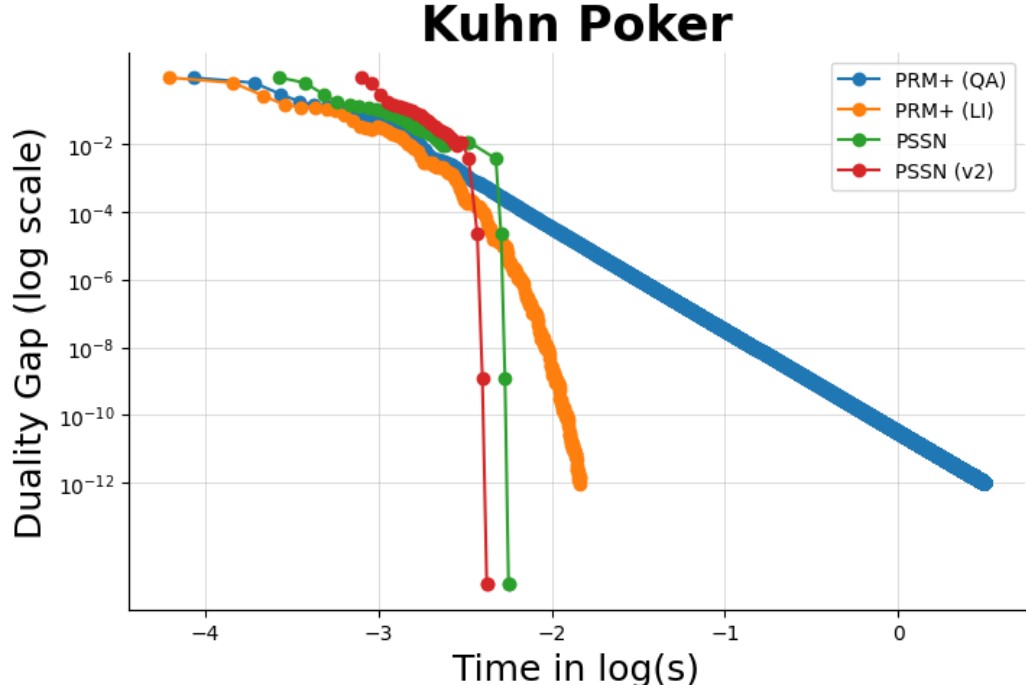

Kuhn poker is a standard benchmark in the game-theoretic literature and is widely used as a baseline for evaluating algorithms. However, its small size makes it straightforward to solve exactly. Even so, we observe improvements with our PSSN methods, which switch at a loose threshold of $10^{-2}$, over the last-iterate PRM$^+$ algorithm, and our methods also outperform PRM$^+$ with quadratic averaging and alternation.

**Layered graph security games** (Černỳ et al., 2024). We test on three classes of layered graph security games: the pursuit-evasion, logical-interdiction, and anti-terrorism games. Detailed descriptions of these instances can be found in Černỳ et al. (2024).

For all layered graph security game tests, we use the random seed 2025 together with the default parameter settings from the associated GitHub repository. The payoff matrix for the pursuit-evasion game has dimension $119 \times 887$ with 70253 nonzeros, the payoff matrix for the logical-interdiction game has dimension $119 \times 856$ with 2931 nonzeros, and the payoff matrix for the anti-terrorism game has dimension $119 \times 1080$ with 3756 nonzeros.

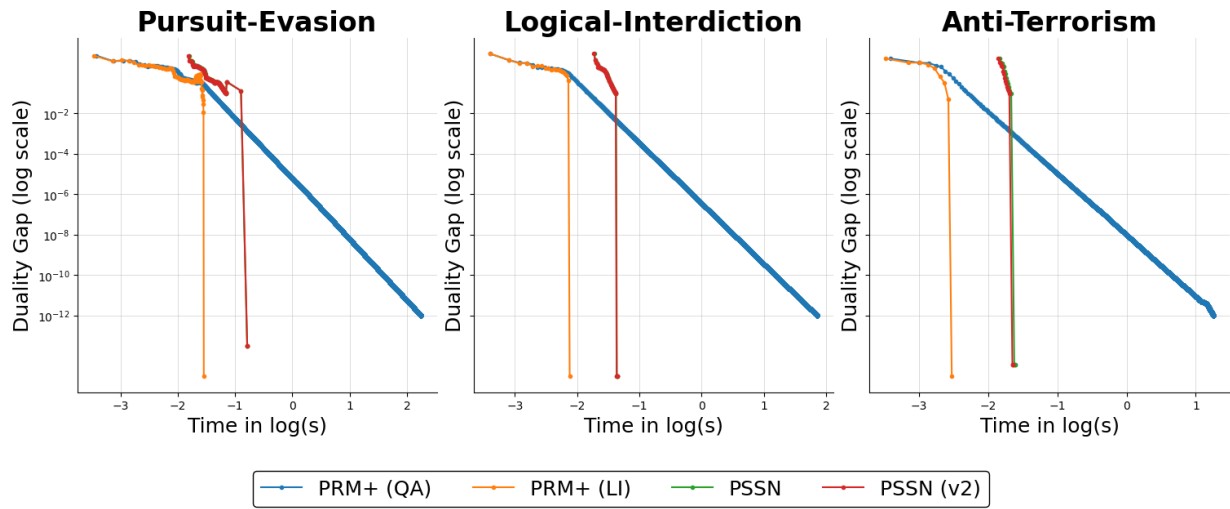

In these instances, the last-iterate PRM$^+$ algorithm solves the game directly and our PSSN methods solve the game quickly after switching at a relatively loose threshold of $10^{-1}$. We note that these instances clearly do not require the use of curvature information to find an efficient solution. However, our results still outperform PRM$^+$ with quadratic averaging and alternation, the de facto approach to game solving. This motivates our focus on random matrix games in our experiments; in random matrix instances, PRM$^+$ struggles to efficiently converge to a high-precision solution, and we find that our PSSN algorithms can do so extremely effectively.

## C.2. Random Game Instances

The following tests are run on 10 independently generated random seeds across two classes of random matrices. In the uniform case, each entry $A_{ij}$ is drawn uniformly from $[-1, 1]$, while in the normal case, each entry $A_{ij}$ is drawn i.i.d from a standard normal distribution. We generate these instances on matrices of size $n \times m$ for $(n, m) = (100, 100), (400, 400)$, and $(400, 800)$.

We first test our methods across different duality gap switching thresholds. The best thresholds, in terms of lowest average clock times, are colored in blue.

*Table 4.* Averaged clock times (seconds) for $100 \times 100$ random matrix games vs thresholds.

*Random Uniform*

| Method | Duality Gap Switching Threshold | | | |
|---|---|---|---|---|
| | $10^{-1}$ | $10^{-2}$ | $10^{-3}$ | $10^{-4}$ |
| PSSN v1 | 0.088 | 0.100 | 0.113 | 0.180 |
| PSSN v2 | 0.083 | 0.098 | 0.111 | 0.180 |

*Random Normal*

| Method | Duality Gap Switching Threshold | | | |
|---|---|---|---|---|
| | $10^{-1}$ | $10^{-2}$ | $10^{-3}$ | $10^{-4}$ |
| PSSN v1 | 0.118 | 0.106 | 0.123 | 0.182 |
| PSSN v2 | 0.116 | 0.105 | 0.120 | 0.186 |

*Table 5.* Averaged clock times (seconds) for $400 \times 400$ random matrix games vs thresholds.

*Random Uniform*

| Method | Duality Gap Switching Threshold | | | |
|---|---|---|---|---|
| | $10^{-4}$ | $10^{-5}$ | $10^{-6}$ | $10^{-7}$ |
| PSSN v1 | 3.154 | 2.645 | 4.213 | 6.866 |
| PSSN v2 | 2.939 | 2.424 | 17.269 | 23.693 |

*Random Normal*

| Method | Duality Gap Switching Threshold | | | |
|---|---|---|---|---|
| | $10^{-4}$ | $10^{-5}$ | $10^{-6}$ | $10^{-7}$ |
| PSSN v1 | 3.762 | 2.645 | 4.519 | 7.132 |
| PSSN v2 | 3.446 | 3.068 | 26.445 | 39.878 |

*Table 6.* Averaged clock times (seconds) for $400 \times 800$ random matrix games vs thresholds.

*Random Uniform*

| Method | **Duality Gap Switching Threshold** | | | |
| | $10^{-4}$ | $10^{-5}$ | $10^{-6}$ | $10^{-7}$ |
|---|---|---|---|---|
| PSSN v1 | 9.070 | 5.940 | 4.648 | 6.839 |
| PSSN v2 | 9.578 | 4.323 | 5.125 | 74.38 |

*Random Normal*

| Method | **Duality Gap Switching Threshold** | | | |
| | $10^{-4}$ | $10^{-5}$ | $10^{-6}$ | $10^{-7}$ |
|---|---|---|---|---|
| PSSN v1 | 35.320 | 5.306 | 33.993 | 10.827 |
| PSSN v2 | 34.497 | 5.094 | 89.87 | 119.25 |

Across sizes and distributions, the best switching threshold is not universal and depends heavily on matrix dimension and problem structure. We note that smaller, better-conditioned matrix games (such as the randomly generated $100 \times 100$ instances) benefit from earlier or immediate switching, while larger, more ill-conditioned matrix games (such as the randomly generated $400 \times 400$ and $400 \times 800$ instances) require tighter thresholds for switching to the SSN phase. At the same time, we also note the tradeoff implicit in updates of the damping parameter: updating too aggressively can destabilize the SSN phase, whereas setting the threshold too tightly can delay the switch, causing $PRM^+$ to run excessively long before the benefits of second-order acceleration are realized.

*Table 7.* Averaged clock times (seconds) for $100 \times 100$ random matrix games.

*Random Normal*

| Method | **Time to Reach Duality Gap Tolerance** | | | | | |
| | $10^{-2}$ | $10^{-4}$ | $10^{-6}$ | $10^{-8}$ | $10^{-10}$ | $10^{-12}$ |
|---|---|---|---|---|---|---|
| $PRM^+$ (LI) | 0.008 | 0.120 | 4.476 | 31.205 | | |
| $PRM^+$ (QA) | 0.008 | 0.050 | 0.471 | 6.238 | 26.312 | |
| PSSN v1 | 0.035 | 0.105 | 0.116 | 0.117 | 0.118 | 0.118 |
| PSSN v2 | 0.035 | 0.102 | 0.114 | 0.115 | 0.115 | 0.116 |

*Random Uniform*

| Method | **Time to Reach Duality Gap Tolerance** | | | | | |
| | $10^{-2}$ | $10^{-4}$ | $10^{-6}$ | $10^{-8}$ | $10^{-10}$ | $10^{-12}$ |
|---|---|---|---|---|---|---|
| $PRM^+$ (LI) | 0.009 | 0.077 | 5.170 | 30.808 | | |
| $PRM^+$ (QA) | 0.008 | 0.041 | 0.396 | 8.452 | 27.946 | |
| PSSN v1 | 0.022 | 0.070 | 0.086 | 0.087 | 0.088 | 0.088 |
| PSSN v2 | 0.020 | 0.066 | 0.081 | 0.082 | 0.083 | 0.083 |

*Table 8.* Averaged clock times (seconds) for $400 \times 400$ random matrix games.

*Random Normal*

| Method | **Time to Reach Duality Gap Tolerance** | | | | | |
| | $10^{-2}$ | $10^{-4}$ | $10^{-6}$ | $10^{-8}$ | $10^{-10}$ | $10^{-12}$ |
|---|---|---|---|---|---|---|
| $PRM^+$ (LI) | 0.005 | 0.088 | 3.140 | 22.092 | | |
| $PRM^+$ (QA) | 0.005 | 0.034 | 0.332 | 4.258 | 18.042 | |
| PSSN v1 | 0.075 | 0.463 | 2.608 | 2.631 | 2.641 | 2.645 |
| PSSN v2 | 0.077 | 0.481 | 2.972 | 3.052 | 3.064 | 3.068 |

*Random Uniform*

| Method | **Time to Reach Duality Gap Tolerance** | | | | | |
| | $10^{-2}$ | $10^{-4}$ | $10^{-6}$ | $10^{-8}$ | $10^{-10}$ | $10^{-12}$ |
|---|---|---|---|---|---|---|
| $PRM^+$ (LI) | 0.006 | 0.054 | 3.608 | 20.130 | | |
| $PRM^+$ (QA) | 0.005 | 0.027 | 0.268 | 5.755 | 18.877 | |
| PSSN v1 | 0.075 | 0.375 | 2.538 | 2.629 | 2.640 | 2.645 |
| PSSN v2 | 0.075 | 0.375 | 2.319 | 2.410 | 2.417 | 2.424 |

In Table 7 and Table 8, we report average clock times across 10 independent random uniform and normal matrix games

of sizes $100 \times 100$ and $400 \times 400$, respectively. For the smaller $100 \times 100$ instances, results are shown with a switching threshold of $10^{-1}$, which reflects the fact that such problems are well-conditioned and allow earlier switching into the DRSSN phase without loss of stability. In contrast, the results for the larger, more ill-conditioned $400 \times 400$ instances are shown with a tighter switching threshold of $10^{-5}$ to ensure that the DRSSN method is initialized within a local convergence regime. These thresholds correspond to the optimal switching values identified in Table 4 and Table 5, and are used here to provide an accurate performance comparison. Again, the results across these two tables highlight how the problem size and conditioning affect the choice of switching threshold, and illustrate the robustness of our hybrid method in both moderate and large-scale random matrix settings.

## D. LLM Usage

The authors used ChatGPT to aid/polish writing.

