# OpenReview forum: "A Direct Second-Order Method for Solving Two-Player Zero-Sum Games"
_ICML.cc/2026/Conference — ICML 2026 regular_

### Official Review · Reviewer_7osg · 2026-02-25

**Soundness:** 3
**Presentation:** 3
**Significance:** 3
**Originality:** 3
**Overall Recommendation:** 5
**Confidence:** 4

**Summary:**

The paper studies iterative methods for computing Nash equilibria in two-player zero-sum games. The authors design a semi-smooth Newton method which achieves local quadratic convergence, when applied to a suitably constructed operator. The method's theoretical foundation rests on a connection via projection between fixed points of the operator and Nash equilibria. Furthermore, this connection allows the use of first-order methods to warm-start the algorithm, i.e. sufficiently reduce the duality gap, until the area of local convergence is reached. The paper includes experiments (using PRM+ as a warm start method and a comparison benchmark) that corroborate the theoretical results.

**Compliance With Llm Reviewing Policy:**

Affirmed.

**Final Justification:**

The authors' rebuttal addressed my concern about scalability by providing more experiments and some directions for future work on the topic.

**Key Questions For Authors:**

Can the authors provide experiments with more strategies (a few thousands) per player?

**Limitations:**

yes

**Strengths And Weaknesses:**

Strengths:
Solving BLSPs is a fundamental topic for the game theory/optimization community.  Designing scalable algorithms for such problems is an active area of research, where second-order methods have not been explored in depth; thus the core contribution of the paper appears important and novel. Moreover, hybrid methods are commonly used in practice, but are relatively unexplored theoretically. Finally, the paper is well written and easy to follow.

Weaknesses:
The lack of a theoretical switching rule, as noted by the authors, is a drawback of the method. This is addressed by the discussion on instance-dependent parameters at the end of Section 3, and in extension by the algorithm’s performance in practice.
The only concern I have is if the advantage of the method in wall-clock time will be preserved in higher dimensional games. While I do not think that the dimensions selected in the paper (400x800) are toy examples, they do not yet qualify as large-scale instances.

---

> ### Author Rebuttal · Authors · 2026-03-30
>
> Thank you for taking the time to review our paper. We provide responses to your comment below:
>
> > "The only concern I have is if the advantage of the method in wall-clock time will be preserved in higher dimensional games. While I do not think that the dimensions selected in the paper (400x800) are toy examples, they do not yet qualify as large-scale instances. Can the authors provide experiments with more strategies (a few thousands) per player?"
>
> We thank the reviewer for this point. We agree that understanding scalability to larger dimensions is important.
>
> In response to your comment, we conducted additional experiments on larger instances  (e.g., 1000×1000 matrix games). We observe that the wall-clock advantage of the hybrid method becomes less pronounced as the problem size grows. In particular, the cost of the SSN phase—primarily due to linear system solves—scales with the game dimension. That said, even in these larger instances, the hybrid method continues to outperform PRM+ in its capability to reach high-accuracy solutions. Furthermore, we see that once PSSN reaches an iterate at $10^{-8}$ duality gap, it progresses to one at $10^{-12}$ in less than one second on average, indicating that after entering the local regime, high-accuracy solutions can be attained in relatively little additional time.
>
> *Table 1: Averaged clock times (in seconds) for 1000 × 1000 random uniform matrix games*
>
> **Time to Reach Duality Gap Tolerance**
>
> | Method    | $10^{-2}$ | $10^{-4}$ | $10^{-6}$ | $10^{-8}$ | $10^{-10}$ | $10^{-12}$ |
> |-----------|-------|-------|-------|-------|--------|--------|
> | PRM$^+$ (QA)  | 0.013 | 0.059 | 0.596 | 7.957 | ($\dagger$)    | ($\ast$)    |
> | PSSN      | 0.218 | 0.727 | 7.193 | 19.020 | 20.433 | 20.605 |
>
>
> Across the 10 random seeds we tested, the symbol $(\dagger)$ indicates that six seeds failed to reach the duality gap threshold, and the symbol $(\ast)$ indicates that all the seeds failed to reach the duality gap threshold. Here, PRM$^+$ is run for 500000 iterations (totaling roughly 60 seconds of wall clock time).
>
> We view scalability as an important direction for further development. The present work should be seen as a foundational step: it demonstrates that second-order information can be effectively incorporated into game-solving pipelines and can significantly accelerate finding high-accuracy solutions.
>
> To further improve performance at larger scales, we may need to modify the algorithm (for example, to incorporate inexact linear system solves rather than exact) or use better heuristics, such as more refined switching and more refined damping parameter updates. We feel these changes give our approach the potential to be scaled to high-dimensional problems. We will revise the paper to better clarify the current scope of the method and its limitations and outline these directions as important future work.

---

> > ### Author Rebuttal · Reviewer_7osg · 2026-04-03
> >
> > I thank the authors for conducting additional experiments as I requested. My takeaway from the provided results is positive, especially the observed drop from $10^{-8}$ to $10^{-12}$. I also find interesting the proposed modifications to achieve scalability (particularly the use of an inexact solver; I think that gradually moving from an approximate solver to an exact as the duality gap decreases may be helpful).
> >
> > Therefore, I maintain my positive opinion about the paper and my score.

---

> > > ### Author Response · Authors · 2026-04-08
> > >
> > > Thank you for taking the time to review our paper! We appreciate the discussion.

---

### Official Review · Reviewer_Lf6e · 2026-03-01

**Soundness:** 2
**Presentation:** 3
**Significance:** 1
**Originality:** 3
**Overall Recommendation:** 4
**Confidence:** 3

**Summary:**

This paper introduces a second-order algorithm to approximately solve two player zero-sum games. The algorithm is based on a Douglas-Rachford splitting formulation which is solved at each iteration using a regularized semi-smooth Newton method. This allows the algorithm to achieve quadratic convergence locally to Nash equilibria, a result which relies on a connection between (approximate) equilibria and fixed points of a appropriate DRS operator. Then, the paper combines PRM+ with the proposed methods to introduce an end to end method, where PRM+ is used to converge quickly to a local neighborhood of a Nash, and the second order method is used to guarantee fast (last iterate) convergence thereafter. Finally, a few empirical comparisons are made to standard first order methods on some benchmark games and random games. Overall, the convergence time of the proposed methods are significantly lower than PRM+, EG and OGDA in randomly generated games.

**Compliance With Llm Reviewing Policy:**

Affirmed.

**Final Justification:**

My main concern with the paper was the lack of empirical justification beyond relatively small random games, and relying on a potentially expensive linear system solving step. The authors provided some clarification and also ran additional experiments, and so have partially addressed my concern. I am still somewhat borderline on this paper because I feel it lacks a truly challenging experiment like a poker-type setting or security game, absent which the claims of superiority of the method are validated only on random games.

**Key Questions For Authors:**

- The convergence of the hybrid procedure relies on an instance dependent parameter $\epsilon(A,\gamma)$. Is there an upper bound on this parameter in the games you study? The concern here is that since Thm 3.9 holds for a subsequence of iterates, in the worst case it could be that even with the PRM+ warm start, the DRSSN procedure could still take a long clock time before entering the quadratic regime. Perhaps I have misunderstood the theorem statements here, so clarifications would be helpful.

**Limitations:**

yes

**Strengths And Weaknesses:**

### Strengths
- The paper is well written and structured. The technical material is clearly presented, and the overall motivation of incorporating second order methods to alleviate potentially slow last iterate convergence is well argued for.
- Technically, the use of second order method to improve convergence is nice idea, and the overall theoretical contribution is meaningful going forward, since it allows for future work exploring hybrid methods that outperform FOMs and LP based methods.


### Weaknesses
- The primary weakness in this paper to me is that the experiments are lacking: games based on applications (like security games) do not result in significant performance loss for PRM+, so the results for random games are chosen instead. The lack of a clear 'realistic' setting where PRM+ performs poorly but the proposed method performs well diminishes somewhat the contribution. Moreover, it is not clear to me why the random game structure necessitates low duality gap tolerance. For instance, in a random game, why should I care about an equilibrium that has duality gap of 10^-12 when I can just run PRM+ which is easier to implement and can give me a 10^-6 solution in a comparable amount of time?
- The authors argue that LPs are not good benchmarks in large/realistic since they require solutions to linear systems, yet the second order method also needs to solve a linear system at each iteration. It would be interesting to see how an LP solver performs on the current experiments and if it is comparable.

As it stands, though the use of second-order methods in game-solving is certainly a conceptually interesting direction, the paper in its current form does not sufficiently argue for the merits of the approach beyond existing methods. Other uses of second-order methods in the literature (see e.g. [1]) have successfully shown convergence in games where efficient algorithms did not exist at all. For the proposed technique to be significantly useful to the community, in my opinion there needs to be a clear subclass or application where high precision Nash are important, and FOMs are inadequate. If I could be convinced of this, then I am happy to increase my score to an Accept.

[1] Daskalakis, Constantinos, et al. "Stay-on-the-ridge: Guaranteed convergence to local minimax equilibrium in nonconvex-nonconcave games." The Thirty Sixth Annual Conference on Learning Theory. PMLR, 2023.

---

> ### Author Rebuttal · Authors · 2026-03-30
>
> Thank you for taking the time to review our paper. We provide responses to your questions and comments below:
>
> > Regarding the merits of our approach beyond existing methods, and relevant application areas
>
> We hope to clarify the broader motivation and positioning of our work. First, second-order approaches to game solving are largely underexplored, despite prior work in more general (nonconvex-nonconcave) settings (e.g. your reference [1]) highlighting limitations of first-order methods. This suggests that incorporating curvature information is important for improving convergence, and we are the first to do so in the classical two-player zero-sum game setting.
>
> High-precision equilibria are especially important in deployment settings, where the goal is to play a strategy closest to the true optimal mixed strategy; note that strategy profiles with only a moderately small gap may still deviate meaningfully from the true optimal strategy. A concrete example is heads-up limit Texas hold’em [2], where, even with substantial computational resources, an exact equilibrium could not be computed, and a statistical notion of “weakly solving” was required. This highlights that achieving very high accuracy solutions is both difficult and practically relevant.
>
> Moreover, even in simplified games, first-order methods can struggle to reach high-precision equilibria! In [3], predictive CFR and its variants struggle in poker-inspired domains like Goofspiel, the River endgame, and Leduc poker. This indicates that first-order approaches alone may be insufficient in the high-accuracy regime.
>
> Our method is designed precisely for this. We use PRM+ as a warm start and introduce a second-order phase that exploits curvature near equilibrium, enabling much faster convergence once in the local region. We view this as a foundational step, with the goal of extending these ideas to extensive-form games in future work.
>
> [2] Michael Bowling, Neil Burch, Michael Johanson, and Oskari Tammelin. Heads-up Limit Hold’em Poker is Solved. Science 2015.
>
> [3] Gabriele Farina, Christian Kroer, and Tuomas Sandholm. Faster game solving via predictive blackwell approachability: Connecting regret matching and mirror descent. AAAI 2021.
>
> > Regarding the purpose of a smaller duality gap solution, and our random matrix game context
>
> We do not claim that our hybrid method universally outperforms first-order methods like PRM+, which can be preferable for low-precision solutions. Rather, our contribution is to enable efficient computation for the high-accuracy regime, where PRM+ can struggle. We emphasize that we use random matrix games in our experiments to provide a controlled setting for studying our method's behavior and to effectively evaluate the theoretical benefits of our hybrid approach.
>
>
> > On comparison of LP iterations and second-order iterations
>
> We will clarify our comparison to LPs in our revision. Our point is not that LP/interior-point methods are expensive because they solve linear systems, whereas our SSN method does not.
>
> Our focus in this work is to develop a hybrid method that exploits the BLSP structure while incorporating curvature information. More broadly, our goal is to build on this framework to design methods that retain these benefits without relying on exact linear system solves—for example, with inexact or iterative solvers such as conjugate gradient, together with the corresponding theoretical and empirical analysis. Notably, even with exact solves, our experiments demonstrate clear advantages over PRM+ in achieving high-accuracy solutions.
>
> As we mentioned in our experiments section, we agree that LP solvers would likely perform comparably on the instances considered in our experiments. However, our intent is to demonstrate that our PSSN methods can outperform PRM+ and to take a first step toward incorporating second-order information into scalable game-solving algorithms.
>
> > On the parameter $\epsilon(A, \gamma)$) and DRSSN
>
> We first clarify that Theorem 3.9 applies to a contiguous subsequence: once DRSSN enters the local neighborhood, all subsequent iterates enjoy quadratic convergence.
>
> We understand that the main question lies in determining the size of the local neighborhood, governed by $\epsilon(A, \gamma)$. We do not have a computable bound on $\epsilon(A, \gamma)$; however, this is consistent with other instance-dependent parameters arising in local convergence analyses (condition measure and Hoffman constants), which are typically unavailable in closed form even for specific problem classes.
>
> Finally, while our theory does not rule out DRSSN taking a long clock time before entering the quadratic regime, this behavior is not observed empirically. In our experiments, switching at a duality gap of around $10^{-5}$ typically places the iterate directly in (or very near) the local region.
>
> We hope we have addressed your concerns and look forward to further discussions.

---

> > ### Author Rebuttal · Reviewer_Lf6e · 2026-04-04
> >
> > Thank you to the authors for the rebuttal and apologies for the delayed response. There is no doubt that first order methods do not find equilibria with high-precision, and the authors' response points towards EFG settings with imperfect information. I think the paper would be significantly improved with additional experiments *beyond random games*, as I do agree with the authors that deployment settings are the most interesting/relevant for this class. In lieu of this, the authors have also run experiments on larger random games, comparing to PRM+. This is reasonably convincing, and I would be happy to increase my score to a borderline accept. Framing the current version as a foundational work seems reasonable to me,  but my core feedback is that it would be crucial to see a 'deployment' type experiment in a concrete game structure (e.g. poker or attacker-defender games) in future versions.

---

> > > ### Author Response · Authors · 2026-04-08
> > >
> > > Thank you for your follow-up and for taking the time to review our paper.
> > >
> > > While we do include a brief experiment on security games in the appendix, we agree that further evaluation in deployment settings where PRM+ struggles (e.g. poker or other security-based games) is an important direction for future work.
> > >
> > > Thank you for the discussion and for raising your score!

---

### Official Review · Reviewer_yvNN · 2026-03-13

**Soundness:** 3
**Presentation:** 3
**Significance:** 3
**Originality:** 3
**Overall Recommendation:** 5
**Confidence:** 2

**Summary:**

This paper studies the problem of computing Nash equilibria in two-player zero-sum games. As stated in the abstract, the main contribution is a second-order method tailored to this setting. Since state-of-the-art first-order methods such as Predictive Regret Matching+ provide global convergence guarantees but may become slow in the high-accuracy regime, the paper proposes to combine such a first-order procedure with a local second-order method that enjoys superlinear convergence once sufficiently close to a solution.

**Compliance With Llm Reviewing Policy:**

Affirmed.

**Final Justification:**

I feel positively about the paper, although I have low confidence in my review.

**Key Questions For Authors:**

- The paper is motivated in part by the goal of handling large games, where LP-based approaches are often computationally prohibitive. However, second-order methods also come with a nontrivial overhead, since each step may require solving a linear system whose size scales with the game. At the same time, one advantage of LP formulations is that they can recover an exact solution, rather than only an approximate one. That said, I would appreciate how the proposed method compares to LP-based approaches in terms of computational complexity.

- In which approximation regime should one expect the proposed method to outperform existing first-order methods? From the experiments, my impression is that the benefits mainly appear in the high-accuracy regime.

- There is also a line of work on predictive/optimistic first-order methods showing local linear convergence to the solution; for example Wei et al. [1]. Of course, such results may depend on constants that are unknown or potentially very large in practice. Still, I wonder whether the authors consider this line of work relevant to discuss more explicitly.

[1] Linear last-iterate convergence in constrained saddle-point optimization CY Wei, CW Lee, M Zhang, H Luo

**Limitations:**

The paper does not highlight any limitation or societal impact, but I do not see any specific negative societal impact here.

**Strengths And Weaknesses:**

Overall, I found the paper well written. The contribution is easy to identify, the motivation is clear, and the presentation follows a coherent storyline throughout. In particular, the transition from the limitations of first-order methods to the proposed hybrid approach is handled nicely, which makes the paper pleasant to read, with a clear previous work overall. Also, the proofs that I checked also seemed correct, but I have not checked everything in great details. That said, I have not identified a major weakness that I should point out. During the rebuttal phase I will probably re-read the paper, and come back with further notes. Instead, I have a few comments and questions that I believe would benefit from further discussion with the authors.

---

> ### Author Rebuttal · Authors · 2026-03-30
>
> Thank you for taking the time to review our paper. We provide responses to your questions below:
>
> > "The paper is motivated in part by the goal of handling large games, where LP-based approaches are often computationally prohibitive. However, second-order methods also come with a nontrivial overhead, since each step may require solving a linear system whose size scales with the game. At the same time, one advantage of LP formulations is that they can recover an exact solution, rather than only an approximate one. That said, I would appreciate how the proposed method compares to LP-based approaches in terms of computational complexity."
>
> We hope to make our positioning clearer in our revision. Our point is not that LP/interior-point methods are expensive because they solve linear systems, whereas our SSN method does not. In terms of computational complexity, we view one SSN step as comparable to one interior-point iteration, since both are dominated by solving a linear system.
>
> Our focus in this work is to develop a hybrid method that exploits the BLSP structure while incorporating curvature information. In doing so, we introduce a direct second-order framework with strong local convergence properties, which we view as a foundation for future algorithmic improvements. More broadly, our goal is to build on this framework to design methods that retain these benefits without relying on exact linear system solves—for example, with inexact or iterative solvers such as conjugate gradient, together with the corresponding theoretical and empirical analysis. Notably, even with exact solves, our experiments demonstrate clear advantages in achieving high-accuracy solutions compared to the state-of-the-art first-order method PRM+.
>
> We will revise the paper to better clarify the current scope of the method and outline these directions as important future work.
>
> > “In which approximation regime should one expect the proposed method to outperform existing first-order methods? From the experiments, my impression is that the benefits mainly appear in the high-accuracy regime.”
>
> Yes, our method is intended for the high-accuracy regimes. As noted in the paper (page 2, second column), in low-precision regimes, our hybrid method can be slower than pure FOMs due to preprocessing steps such as computing the required resolvents. In the context of game solving, however, high-accuracy solutions are especially important; the goal is to obtain a strategy close to the true optimal mixed strategy, and a strategy profile with only a moderately small gap may still deviate meaningfully in practice from that optimum.
>
> > “There is also a line of work on predictive/optimistic first-order methods showing local linear convergence to the solution; for example Wei et al. [1]. Of course, such results may depend on constants that are unknown or potentially very large in practice. Still, I wonder whether the authors consider this line of work relevant to discuss more explicitly.”
>
> We do think that the line of work on predictive/optimistic first-order methods, including Wei et al. [1], could be relevant and discussed more explicitly. We cite this paper in our convergence analysis (as we build on the metric subregularity property discussed there), but we can clarify this connection further in the revision. Broadly speaking, classical PRM+ admits sublinear guarantees, while some predictive/optimistic variants achieve local linear convergence. At the same time, we do think that local linear convergence in FOMs is a fundamentally distinct topic from locally linear (or superlinear) convergence in second-order methods. For FOMs, these local linear convergence rates are quite poorly understood, and it is not so clear that they are practically meaningful. They often occur after quite a lot of iterations, and are not guaranteed for the most practical methods for solving large games.
>
> A key innovation of our work thus lies in achieving local quadratic convergence of the SSN phase. Our hybrid method thus combines the strong empirical performance of PRM+ as a warm-start with a second-order phase that effectively exploits curvature information near equilibrium. This leads to speedups in both theory and practice.

---

> > ### Author Rebuttal · Reviewer_yvNN · 2026-04-02
> >
> > Thank you for the clarification on the questions.

---

> > > ### Author Response · Authors · 2026-04-08
> > >
> > > Thank you for taking the time to review our paper! We appreciate the discussion.

---

### Decision · Program_Chairs · 2026-04-30

**Decision:**

Accept (regular)

**Comment:**

This work develops the first direct second order method for Nash equilibria in two-player, zero-sum games. The reviewers generally appreciated the novel approach and found the paper well written. While the proposed algorithm’s (Hybrid SSN) primary empirical advantage beyond FOM’s is in finding very high precision solutions, the second order approach is qualitatively distinct from prior work and should be disseminated to the community. Lastly, I think it could be helpful to briefly clarify to a more general audience the meaning of “direct” in a sentence by differentiating from prior algorithms that simply use second order information to improve an iterative update direction, e.g., the general iterative scheme of Dafermos and modern variants [1, 2].

[1] Dafermos, S. An iterative scheme for variational inequalities. Mathematical Programming. 1983.

[2] Schäfer, F & Anandkumar, A. Competitive Gradient Descent. NeurIPS. 2019.